# First-Year Survival of Northern Fur Seals (*Callorhinus ursinus*) Can Be Explained by Pollock (*Gadus chalcogrammus*) Catches in the Eastern Bering Sea

Jeffrey W. Short [1,*], Harold J. Geiger [2], Lowell W. Fritz [3] and Jonathan J. Warrenchuk [4]

[1] JWS Consulting LLC, 19315 Glacier Highway, Juneau, AK 99801, USA
[2] St. Hubert Research Group, 222 Seward, Suite 205, Juneau, AK 99801, USA; geiger@ak.net
[3] 2532 234th Place SW, Brier, WA 98036, USA; alfritz2532@gmail.com
[4] Oceana, Inc., 175 S. Franklin St. #418, Juneau, AK 99801, USA; jwarrenchuk@oceana.org
[*] Correspondence: jwsosc@gmail.com

**Abstract:** The Pribilof northern fur seal (*Callorhinus ursinus*) herd in the eastern Bering Sea has declined by ~70% since the 1970s, for elusive reasons. Competition for pollock (*Gadus chalcogramma*) with the commercial fishery has been suspected as a contributing factor, but no correlative relationship between fishing activity and fur seal population declines has heretofore been demonstrated. Here, we present evidence for a moderately strong inverse relationship between fishery catches of pollock and first-year survival of fur seals, based on three different approaches to evaluation. We suspect this relationship results from the dependence of lactating female fur seals on locating dense and extensive schools of pollock near the Pribilof Islands to efficiently provide nutrition for their pups, because the pollock fishery also targets these same schools, and when fished, the remnants of these schools are fragmented and dispersed, making them more difficult for fur seals to locate and exploit. Inadequately fed pups are less likely to survive their initial independent residence at sea as they migrate south from the Pribilof Islands in the fall. Our results imply that pollock catches above ~1,000,000 t within ~300 km of the Pribilof Islands may continue to suppress first-year survival of Pribilof fur seals below the estimated equilibrium survival value of 0.50, leading to continued decline of the population.

**Keywords:** ecosystem-based fishery management; fishery-apex predator competition; forage fish; North Pacific Ocean regime shift

## 1. Introduction

The Pribilof Island breeding population of northern fur seals (*Callorhinus ursinus*) has been in long-term decline since the mid-1950s, after fully recovering from severe over-harvesting in the late 19th and early 20th centuries [1]. The population was designated as depleted by the U.S. National Marine Fisheries Service (NMFS) in 1988 because it had declined to < 50% of the 1950s peak (~1.6 million) without evidence of a change in habitat carrying capacity [2]. The population has continued to decline and was estimated to number less than a half million in 2018 [3,4]. An effort to increase population fecundity by deliberately killing adult females, both on land from 1956 to 1968, and at sea from 1958 to 1974, accounts for most of the decline through the 1970s [5,6]. However, the continuing decline thereafter remains largely unexplained [7–10].

The northern fur seal plays a key role in the marine food web of the eastern Bering Sea (EBS) [11,12]. From May through October each year, fur seals use the Pribilof Islands to breed, bear and nurture their young [1,11,13–17]. Isolation from most terrestrial predators and proximity to the highly productive waters of the EBS has historically provided exceptionally favorable conditions for fur seal growth and survival, particularly for adult females nurturing new-born pups. Consuming up to ~1 million t of fish and squid each

breeding season [12], the high abundance of northern fur seals on the Pribilof Islands placed considerable demands on the productivity of prey populations in adjacent waters. Along with Steller sea lions (*Eumetopias jubatus*) and several piscivorous seabird species that also breed and rear their young contemporaneously with northern fur seals, and have had recent population declines [4,18], these near-apex predators may substantially modulate populations of forage fishes and squids of the EBS.

Proposed causes of the recent decline of the Pribilof fur seal population include pup trauma on land, starvation, disease, increased predation, environmental change, entanglement by derelict fishing gear, mortality from bycatch in commercial fisheries, commercial and subsistence harvest of fur seals, and competition with commercial fisheries for prey [2,7–9]. Adverse weather conditions during the initial southward migration of fur seal pups have also been suggested as a factor [7,19,20]. Since the 1970s, fur seal mortality from commercial fishing bycatch, and commercial and subsistence harvests have been too small to have had a substantive effect at the population level [2]. Population modeling studies suggest that decreased survival of juveniles and adult females resulting from competition with commercial fisheries for prey and increased natural mortality of adult females from entanglement with derelict fishing gear may be important factors [8,21]. Studies of other possible causes have failed to account for the decline [2,10,22].

A recent bioenergetic modeling study suggests that the commercial pollock (*Gadus chalcogramma*) fishery in the EBS may directly compete with northern fur seals for food during the fur seal breeding season [21]. Fur seals prey heavily on pollock in the Bering Sea, especially lactating females nurturing their pups from their birth in early summer through weaning and their departure in the late fall [21]. Simultaneously, the waters surrounding the Pribilof Islands have been intensely fished by the US commercial pollock fishery in the EBS [23]. From 1970 to 2020, an average of 1.2 million t of pollock was caught annually in the EBS, with generally over half caught from early June through October [23]. Decreased availability of pollock to lactating northern fur seals may impair the ability of females to supply their pups with milk for optimal growth before weaning in the fall, thereby decreasing pup's survival probability through the remainder of their first year of life, when their survival is the lowest until late adulthood [5,21] (Table 1).

Despite substantial circumstantial and other evidence implicating competition with the EBS pollock fishery, a meaningful inverse correlation between increased catches and decreased fur seal recruitment has yet to be demonstrated. Direct comparison of pollock catches with Pribilof northern fur seal population estimates do not appear to be related, and the absence of a clear correlation suggests that other causes may account for the fur seal population decline. However, even if high harvests of pollock directly cause population declines through a lowering of survival in the first year of life, this correlation would not be obvious or readily detectable. Because fur seal population assessments are based on estimates of pup births, any change in abundance would be only be measurable in population assessments several years into the future, after the affected age class moved into peak breeding ages and produced offspring [4–6,10]. Additionally, a reduction in age-0 survival during a single year, or a few years clustered in time, would produce a change in the resulting future population measurements that would be attenuated by presence of breeding females from many other age classes that may not have been similarly affected. Fur seal females attain maturity between the ages of 4 and 8 years, and most produce one pup annually for more than a decade thereafter [5]. Consequently, evaluation of the strength of any relationship between pollock catches in the EBS and recruitment to the fur seal population must properly account for the time lag between changes in first-year survival and pup production 4 to 20 years later.

Table 1. Equilibrium life table for female northern fur seals (adapted from [5]).

| Age | Age-Specific Survival | Cumulative Survival | Fecundity [a] | Proportion of Population Fecundity [b] | Cumulative Proportion of Population Fecundity to Age |
|---|---|---|---|---|---|
| 0 | | 1.000 | 0 | 0 | 0 |
| 1 | 0.500 | 0.500 | 0 | 0 | 0 |
| 2 | 0.800 | 0.400 | 0 | 0 | 0 |
| 3 | 0.840 | 0.336 | 0 | 0 | 0 |
| 4 | 0.920 | 0.309 | 0.0205 | 0.0063 | 0.0063 |
| 5 | 0.940 | 0.290 | 0.1855 | 0.0538 | 0.0601 |
| 6 | 0.940 | 0.273 | 0.3500 | 0.0956 | 0.156 |
| 7 | 0.945 | 0.258 | 0.4000 | 0.1032 | 0.259 |
| 8 | 0.950 | 0.245 | 0.4250 | 0.1041 | 0.363 |
| 9 | 0.950 | 0.233 | 0.4350 | 0.1014 | 0.464 |
| 10 | 0.938 | 0.219 | 0.4400 | 0.0964 | 0.561 |
| 11 | 0.924 | 0.202 | 0.4400 | 0.0889 | 0.650 |
| 12 | 0.906 | 0.183 | 0.4400 | 0.0805 | 0.730 |
| 13 | 0.884 | 0.162 | 0.4350 | 0.0705 | 0.801 |
| 14 | 0.858 | 0.139 | 0.4200 | 0.0584 | 0.859 |
| 15 | 0.876 | 0.122 | 0.4050 | 0.0494 | 0.909 |
| 16 | 0.789 | 0.096 | 0.3850 | 0.0370 | 0.946 |
| 17 | 0.743 | 0.071 | 0.3550 | 0.0252 | 0.971 |
| 18 | 0.692 | 0.044 | 0.3150 | 0.0139 | 0.985 |
| 19 | 0.630 | 0.031 | 0.2800 | 0.0087 | 0.993 |
| 20 | 0.564 | 0.017 | 0.2350 | 0.0040 | 0.997 |
| 21 | 0.490 | 0.008 | 0.1850 | 0.0015 | 0.999 |
| 22 | 0.411 | 0.003 | 0.1300 | 0.0004 | 0.999 |
| 23 | 0.330 | 0.001 | 0.0505 | 0.0001 | 1 |
| 24+ | <0.330 | <0.001 | 0 | 0 | |

[a] Based on births of females only. [b] Product of cumulative survival and fecundity.

Our objective here is to evaluate the strength of any relationship between catches of pollock by the EBS commercial fishery and estimates of Pribilof fur seal pup births after explicitly accounting for changes in fur seal vital rates. We evaluate this relationship in three ways: (1) empirical comparisons of cross-correlations between annual EBS pollock catches and total pollock biomass against the estimated number of pup births after lags of one to 15 years; (2) a simple modification of a Leslie matrix population projection of seal pup births that includes functions of pollock catch or biomass which modify the first-year survival probability in the equilibrium Leslie matrix for fur seals; and (3) a search for Bayesian posterior probabilities of first-year pup survival when EBS pollock catches are zero, low, medium or high.

## 2. Methods

### 2.1. Pribilof Fur Seal Habitat, Life History and Data Sources

Pribilof fur seals range throughout the North Pacific Ocean from mid-October until May or June, when most return to the Pribilof Islands (Figure 1). Adult males arrive around the beginning of May, adult females and older juvenile males arrive in June, while juvenile females and younger juvenile males arrive in July. Age-1 seals of both sexes arrive in late

September and October [17], although most age-1 seals remain at sea until ages 2 or 3 [1]. Fur seals established a new rookery in the early 1980s on Bogoslof Island, an active volcano on the southern margin of the Bering Sea about 40 km north of Umnak Island and 210 km southwest of St. George Island (Figure 1). Females that breed on Bogoslof Island and pups that are born there have the same migratory patterns and occupy the same habitats in the late fall, winter, and spring as Pribilof seals [20,24].

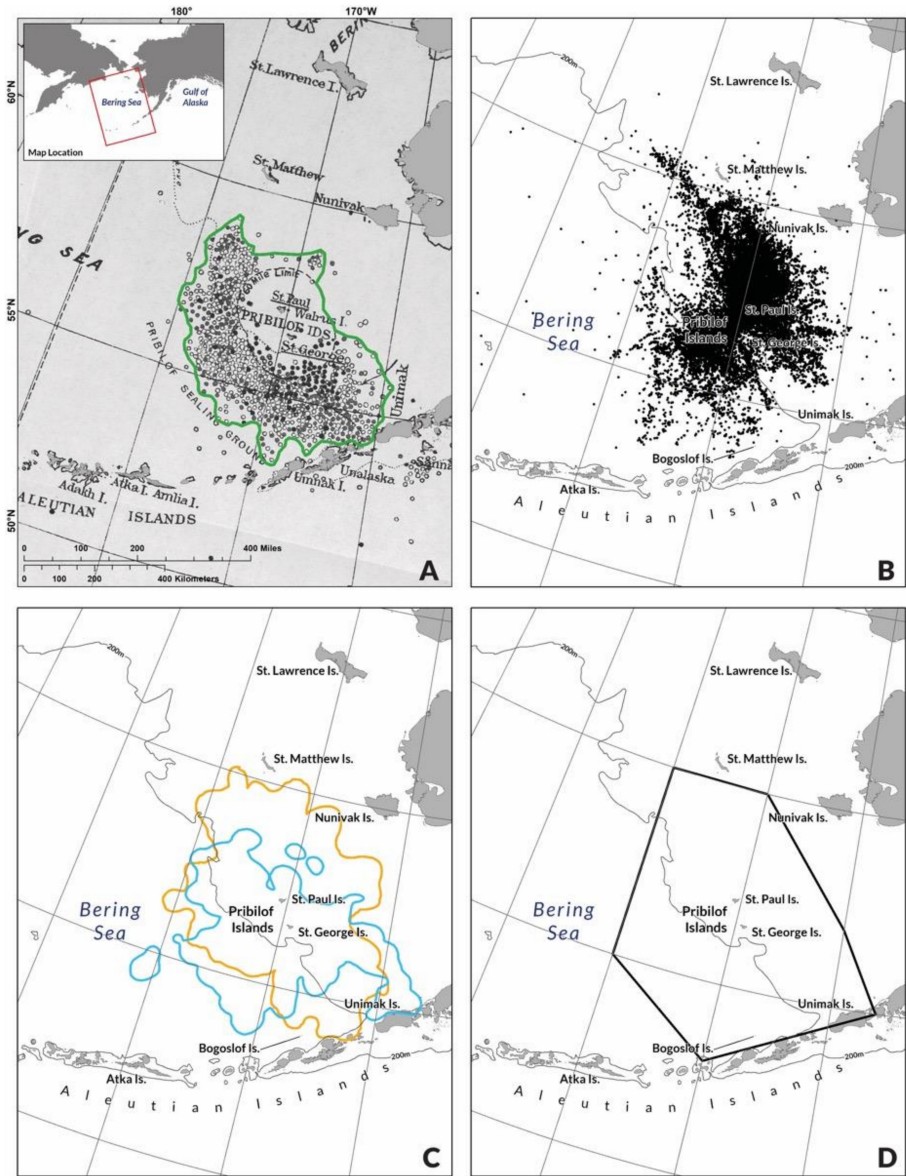

**Figure 1.** (**A**) Pelagic sealing takes in the Pribilof region from 1883 to 1897 based on logbooks from 123 pelagic sealing vessels [25]. Green line encompasses 95% of Bering Sea records. (**B**) Telemetry-derived foraging locations of lactating female fur seals from St. Paul (*n* = 20) in the summer of 2005 and 2006 [26]. (**C**) Meta-home ranges of telemetry-tagged lactating female fur seals from St. Paul and St. George Islands in 1995 and 1996 (yellow boundary; *n* = 97) [27] and St. Paul Island in 2009 (blue boundary; *n* = 82) [28]. (**D**) Simple polygon of female northern fur seal summer foraging habitat that encompassed 95% of data sources (area = 385,300 km$^2$).

Adult females recruit to a harem occupied by a single bull, bear their young within two days of their arrival, are impregnated about 6 days later, and resume foraging at sea the following day [14,15]. Half of annual births occur by mid-July [16]. Harems contain from one to more than a hundred females [14,15], averaging 39 in 1951 [16]. By convention,

nominal age advances one year when an individual fur seal returns to the Pribilof Islands in the summer.

Fur seals migrate to more southerly latitudes of the North Pacific Ocean around the middle of October through November, where they remain until the following spring. Pups are weaned abruptly in the fall and begin their southward migration immediately thereafter [14,29], reaching the Aleutian passes within 11 days on average [30]. Some pups remain within the Bering Sea into January [31]. Pups have little if any experience foraging for food prior to their southward migration [32].

Mature males may attain body weights exceeding 200 kg, while mature females rarely exceed 42 kg to 49 kg, depending on the status of their pregnancy [5]. Males and females mature between ages 7 to 10 and 4 to 8, respectively [5,33]. Male fur seals, mostly between the ages of 1 and 4, were harvested for their pelts almost continuously beginning soon after Russian discovery of the Pribilof Islands in 1786–87 [34] through1984, when the commercial harvest was terminated [2]. In a population at equilibrium, about half of newborn pups die before reaching age 1 [5]. Estimated pup survival on land from birth to weaning averaged 0.901 (range: 0.786–0.974) from 1950 through 1987 [35]. Annual survival rates from ages 2 through 14 years are 0.80 or higher, and males and females rarely live longer than 17 years and 24 years, respectively [5]. Females that survive to age 4 bear 5.8 pups on average over their remaining lifespan [6], rarely bearing more than one pup annually. The sex ratio of pups at birth is approximately 50:50, and more than half of all births occur to females between the ages of 6 to 11 years [5]. Estimated equilibrium female population survival and fecundity rates are listed in Table 1 (from [5]).

The majority of Pribilof fur seals breed and rear their young on St. Paul Island, with fewer on St. George Island and Sea Lion Rock (Figure 1). Increasing numbers of breeding females on Bogoslof Island since the 1980s now exceed those on St. George Island [4]. Fur seal population estimates on each island are based on estimates of the total number of pups born (pup production). Live pup numbers are estimated using a mark-recapture method in August, when early pup mortality during June and July is also estimated. Pup production is used to estimate numbers of older-aged males and females using factors derived from an equilibrium life table [5,36]. Note that this approach to population estimation ensures that post-weaning mortality of female pups and juveniles will only be detectable at a later time through the corresponding reduction in population fecundity resulting from the lost females. For female pups that die during their first year of life, the ensuing reduced fecundity will be most evident after a delay of 6 to 11 years, the ages when females make more than half their total contribution to population fecundity (Table 1).

Northern fur seals opportunistically consume forage fish and squid as available throughout the habitats they occupy during their annual migrations in the North Pacific Ocean [37]. While residing on the Pribilof Islands, most Pribilof fur seals forage within ~400 km of the islands [25,27,28,38] (Figure 1), consuming mainly pollock, gonatid squids and northern smoothtongue (*Leuroglossus schmidti*), with smaller quantities of other gadids, Pacific herring (*Clupea pallasii*), salmonids (*Oncorhynchus sp.*), capelin (*Mallotus villosus*) and Atka mackerel (*Pleurogrammus monopterygius*) also consumed [14,37,39–46]. Adult fur seals may consume both juvenile and adult pollock, and annual variability in the sizes of pollock consumed can be explained by the recruitment history of strong pollock year-classes [44,46–54]. Pollock is the dominant prey species in the EBS overall, especially around the Pribilof Islands and on the EBS shelf. Capelin and Atka mackerel are the main prey consumed near Unimak Pass and in the outer shelf domain north of Unimak Island, respectively, while northern smoothtongue and gonatid squid are the dominant prey in continental slope and oceanic waters [39–44]. Diets vary between rookery islands in the EBS due to the different foraging habitats available to seals breeding on them [27,39–44,50,51,55,56]. The foraging range of seals breeding on St. George includes more off-shelf, basin habitats than St. Paul seals, which lowers the prevalence of pollock in their diet and increases the prevalence of basin and slope prey species (e.g., squid and smoothtongue) [27,50,51]. Fur seals breeding on Bogoslof forage almost exclusively in

basin waters, which results in the lowest prevalence of pollock in fur seal diets of the three rookery islands [26,50,51,55,56].

We base most of our data analysis on St. Paul Island estimates because the majority of Pribilof fur seals breed there, the record of pup counts there is longer and more complete, and the fur seal population estimates for St. Paul and St. George Islands are moderately correlated ($r = 0.77$, $n = 23$). Pup counts were conducted annually on St. Paul Island from 1950 through 1970, and from 1972 through 1990, after which counts were conducted biennially [3,36,47,57,58]. We restrict our analysis to population estimates of females because they determine population fecundity.

### 2.2. Bering Sea Pollock Fishery and Data Sources

Commercial fishing catches in the EBS remained relatively modest until the early 1960s, when implementation of shipboard methods for processing minced fish (surimi) by Japan led to rapidly increasing catches of pollock [59]. Combined catches of pollock and yellowfin sole (*Limanda aspera*), primarily by Japanese fishing vessels, remained below 50,000 t until the mid-1950s, when yellowfin sole catches increased to 554,000 t in 1961, declining abruptly thereafter [60]. Pollock catches increased from 175,000 t in 1964 to 1.87 million t in 1972 (Figure 2B), accounting for about ~80% of the total EBS groundfish catch during this period, with effort occurring in May through September [60]. Total pollock catches have generally ranged between 800,000 t to 1,500,000 t since 1973 (Figure 2B).

We define a Pribilof fur seal foraging area (Figure 1) based on the adult female and juvenile male foraging ranges of Pribilof-based seals as described by [25–28]. We requested and received from NMFS their best documentation of the biomass of pollock observed caught inside and outside the foraging area each year from 1964 through 2018 (Figure 2B). NMFS estimates the biomass and location of pollock catches in the EBS from a combination of logbook entries by vessel captains in the early years of the fishery [61] and from data collected by fishery observers of sampled catches on observed trips (1973 to present) [62]. We scaled spatial catch data inside and outside the foraging area proportionally such that the sum matched the total eastern Bering Sea catch estimates in [23]. While there is some uncertainty in catch amounts and location, particularly in years that had low observer coverage, we made the same assumption as NMFS that fishery catches are estimated without error.

Estimates of total (age 3+) pollock biomass in the entire EBS population, including inside and outside our Pribilof fur seal foraging area, beginning in 1964 are produced each year as part of the stock assessment process, and are presented in Table 29 of [23]). We also computed estimates of total age-1+ biomass, and of the total biomass of ages 1–3, as the product of the weight-at-age averaged over 1991 through 2018 (presented at the bottom of Table 25 of [23]) and the abundance-at-age (presented in Table 30 of [23], summed across ages, for the respective age ranges (i.e., ages 1+ and 1–3).

### 2.3. Effects of Fur Seal Recruitment Perturbations

To show the differences between the kinds of patterns in recruitment that would be observed by a reduction in adult survival versus a reduction in survival of pups, we begin with an evaluation of four hypothetical perturbations to an equilibrium population of Pribilof fur seals. We model these perturbations as changes to the corresponding age-specific survival terms in the equilibrium Leslie matrix for the population for the years when the perturbations are assumed to occur.

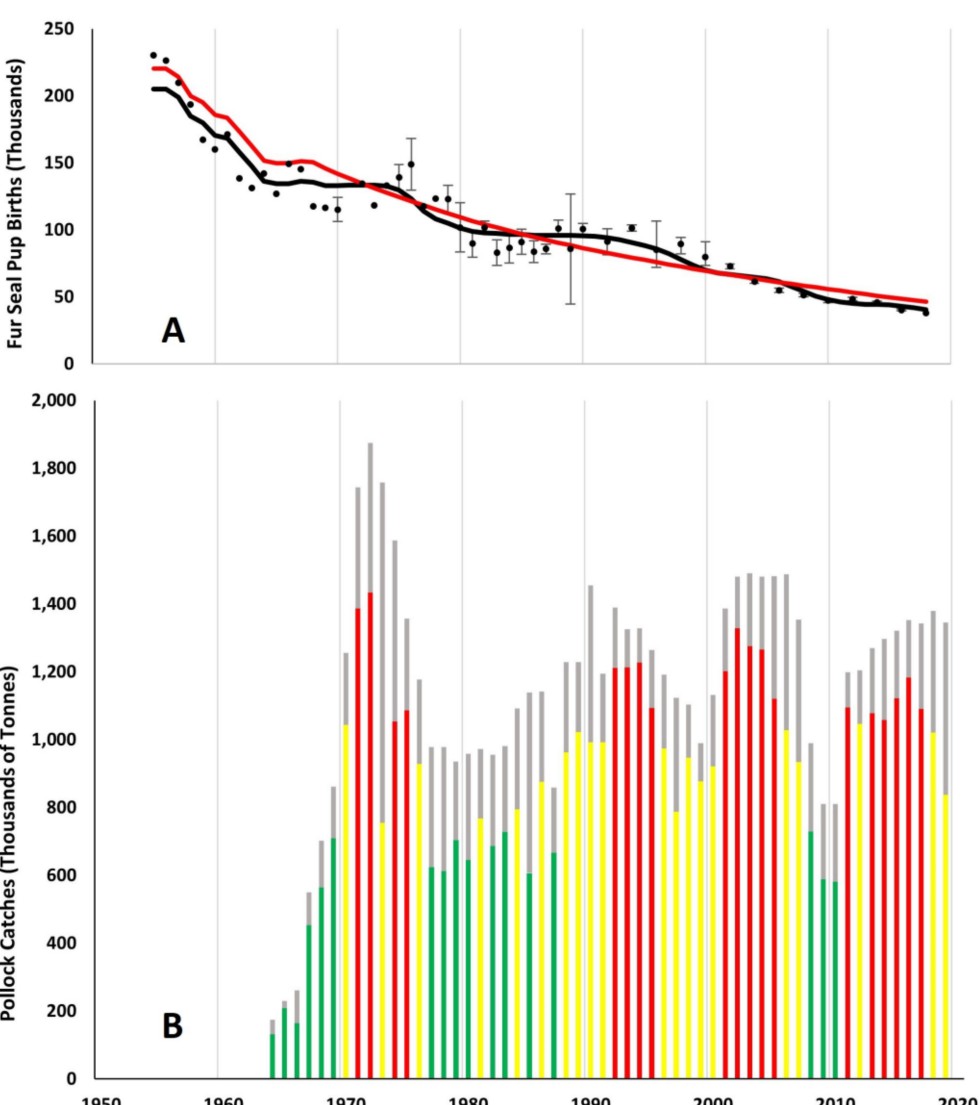

**Figure 2.** (**A**) Observed pup counts on St. Paul Island and predicted pup counts from deterministic models of pup births driven by pollock catch from our Pribilof fur seal foraging area (Model 1, black line) and by estimated age-3+ pollock biomass from the eastern Bering Sea stock assessment [23] (Model 2, red line). (**B**) Annual pollock commercial catches (metric tons) inside our fur seal foraging area during low (green bars), medium (yellow bars), and high (red bars) catch years, and annual total pollock eastern Bering Sea pollock catch (gray bars).

The equilibrium Leslie matrix [58] for the Pribilof fur seal population, which we denote as *A*, is:

$$A = \begin{bmatrix} f_0 & f_1 & & f_{24+} \\ s_0 & 0 & \cdots & 0 \\ 0 & s_1 & & 0 \\ \vdots & & \ddots & \vdots \\ 0 & 0 & \cdots & s_{24+} \end{bmatrix} \tag{1}$$

The elements in the first row of *A* are age-specific fecundities, $f_i$ ($i = 0, 1, 2, \ldots, 24+$, where 24+ includes females aged 24 and above), and the elements on the succeeding rows are age-specific survivals, $s_i$. Lander [5] derived these values from observations of age and pregnancy status of 16,269 female northern fur seals killed at sea during 1958 through 1974 [6], which provide an exceptionally accurate and detailed basis for estimating survival- and fecundity-at-age. Lander [5] assumed the population structure was at equilibrium

during the late 1970's, by which time population estimates appeared to have stabilized and most of the consequences of the female harvest programs had passed through the population (less than 650 females were killed annually after 1968 [6]). We assume here that the population was also at equilibrium during the first half of the 1950s, when the population was at its twentieth-century peak [32].

If $N_y$ is a column vector with 25 elements, with the population abundance at age $i$ in the $i$th element, then the theoretical population change from one year to the next is found by matrix multiplication:

$$N_y = AN_{y-1} \tag{2}$$

Such a theoretical population is at equilibrium if the first eigenvalue of *A* is exactly 1, declining if it is less than 1, or increasing if it is greater than 1 [63].

To evaluate the theoretical effects of perturbations that reduce survival of fur seal adults or pups, we assume that the population has an equilibrium age distribution, found from the first eigenvector from Lander's [5] deterministic Leslie matrix *A* [63]. We scaled this eigenvector so as to produce 200,000 female pups annually. In year six, we reduce the survivals ($s_i$) of all animals in the population by either 10% or 50% for five years, or reduce the survival of pups ($s_0$) by 50% for five years or to zero for one year, computing the number of pups born annually ($N_{0,y}$) throughout from the first to 20 years by extracting the first element of the age distribution vector using Equation (2).

Next, we compare these results with those of a cross-correlation analysis between St. Paul fur seal pups born in year $y$, and pollock catches within our Pribilof fur seal foraging area in year y + h ($-15 \leq h \leq 15$), as an exploratory data analysis tool and as a means of hypothesis generation. Although mechanisms in which pup recruitment would affect pollock catches into the future may appear implausible, independent time series with large serial correlation can appear to be correlated [64]. In this case, the observed auto-correlation in the pollock catch time series ($r = 0.724$) and in the pup abundance time series ($r = 0.710$) are fairly high. It is therefore helpful to examine correlation magnitude in both directions in time as a check to see if the correlation in the plausible direction is larger. In addition, pollock catches might affect pup recruitment into the future if pollock catch somehow reduced the survival of adult females. This would produce a different pattern than if the pollock fishery affected the survival of pups. Similarly, we consider cross-correlations between pups born in year $y$ and pollock biomass in year $y + h$, assuming pollock biomass is linearly related to pollock density. In all cases, the ranges of the time series are from 1964 through 2018.

### 2.4. Pup Survival Perturbation Models

We model effects of pollock catch (Model 1), pollock biomass (Model 2), and catch and biomass acting together (Model 3) on first-year survival of St. Paul fur seals to see how reduced pup survivals could affect the subsequent number of female pups born, using a modified Leslie matrix approach. For this analysis we again assume that the values for fecundity- and survival-at-age presented in Lander [5] accurately determine the equilibrium population structure.

Model 1: We model the effect of pollock catches on first-year survival of St. Paul fur seal pups and subsequent births of female pups by introducing a logistic function of catch that modifies the first-year survival term of the equilibrium Leslie matrix. To evaluate the plausibility that most of any adverse effects on pup survival may occur during the pups' first month of independent life at sea, we partition the age-0 survival term ($s_0$) in the Leslie matrix into three successive intervals: from birth (taken as July 1) to the end of October (denoted as $s_{0,0-4}$); the first month of the pups' southward migration ($s_{0,5}$); and the survival during the last 7 months of their first year of life ($s_{0,6-12}$). We assume that the cumulative survival during their four months on land, $s_{0,0-4}$, is 0.962, the average estimated survival of males during this period for the years 1980–1987, when commercial harvests of fur seals on the Pribilof Islands had largely subsided [31]. We assume the annualized survival during the last 7 months of the first year of life is the same as the survival during the second

year of life (i.e., $s_{0,6-12} = (s_1)^{7/12} = (0.80)^{7/12} = 0.878$; Table 1). Constraining the product $(s_{0,0-4})(s_{0,5})(s_{0,6-12})$ to equal the equilibrium life table survival for $s_0$ of 0.5 gives $s_{0,5} = 0.592$ for the equilibrium survival during the first month of the pups' southward migration.

For years when pollock were caught by the commercial fishery, we subtract a logistic function of the size of the catch $C_y$ for each year $y$ from $s_{0,5}$, so that annual age-0 survival $s_{0;y}$ in the Leslie matrix for year $y$ is computed as:

$$s_{0;y} = (s_{0,0-4})\left[(s_{0,5})\left(1 - \frac{1}{1 + exp(-k(C_y - C_0))}\right)\right](s_{0,6-12}) \tag{3}$$

We choose a logistic function to model effects of catch to reflect the facts that: (1) reduced availability of pollock that result from catch removals are not likely to increase the survival of fur seal pups; (2) small catches likely have little effect on pup survival, and (3) as catches approach the total available pollock biomass, increased catches also have little effect on pup survival if their survivals approach zero. Regarding (1) above, it has been suggested (e.g., Swartzmann and Haar [65]) that high catches of cannibalistic adult pollock could increase the abundance of juvenile pollock, and thus could be beneficial to fur seals. However, we now know that adult fur seals consume both juvenile and adult pollock (see Discussion Section 4.3). The terms $C_0$ and $k$ in the logistic function determine the size of catches that reduce the equilibrium month 5 survival $s_{0,5}$ by half, and the rate that $s_0$ changes with catch when catches are near $C_0$, respectively. We use the annual estimates of pollock catches, including all commercial fishing seasons within a calendar year, from within our Pribilof fur seal foraging area (Figure 1) from 1964 through 2018 (depicted in Figure 2B) for $C_y$ in Equation (1).

Model 2: We model effects of pollock biomass on first-year pup survival similarly to Model 1, by replacing the logistic function in Equation (3) with an equation describing a type II functional response (*sensu* [66]) of survival to changes in pollock biomass, so that:

$$s_{0;y} = (s_{0,0-4})\left[\left(\frac{aB_y}{1 + ahB_y}\right)\right](s_{0,6-12}) \tag{4}$$

We choose a type II functional response of pup survival to pollock density to reflect rapid increases in survival to increased pollock density when the density is low, approaching an asymptote of $1/h$ when density is high. We use annual estimates of total age 1–3, age 3+, and age 1+ pollock biomass from 1964 through 2018 derived from Tables 25, 29 and 30 in the 2019 EBS pollock stock assessment [23], normalized as a proportion of the average, for $B_y$ in Equation (4). The constants $a$ and $h$ are dimensionless because $B_y$ is, so $a$ and $h$ represent times spent searching for pollock and for processing pollock once located, respectively, relative to these times for the average value of $B_y$. Assuming the area inhabited by the pollock population estimated in the pollock stock assessment remains nearly constant, normalization of $B_y$ serves as a proxy for population density per unit sea surface area. If this assumption is substantially violated, then $B_y$ may be interpreted as an index of relative total biomass.

Model 3: We model combined effects of pollock catch and biomass by combining the expressions for $s_{0,5}$ in Equations (3) and (4), with an added estimated parameter $\pi$ to represent the relative proportional contributions of catch and biomass to changes in $s_{0,5}$, so that:

$$s_{0;y} = (s_{0,0-4})\left[(s_{0,5})\left(\pi\left(1 - \frac{1}{1 + exp(-k(C_y - C_0))}\right) + (1 - \pi)\left(\left(\frac{aB_y}{1 + ahB_y}\right)\right)\right)\right](s_{0,6-12}) \tag{5}$$

For all models, we estimate the initial population of female pups in 1955, and the parameters specific to each model ($C_0$ and $k$ in Equation (3), $a$ and $h$ in Equation (4), and $C_0$, $k$, $a$, $h$ and $\pi$ in Equation (5)) by searching for the values that minimize the sum of squared differences between model results and estimates of pups born on St. Paul Island

from 1955 through 2018, using the "Solver" utility in Microsoft Excel. We subtract the numbers of female fur seals killed by age during the 1956–1974 harvest programs [6] from the results of the population distribution projected by the perturbed Leslie matrix each year to account for the effects of these removals on pup production in future years. We reduced the numbers of females reported killed at sea in [6] by 23% to account for the proportion of these mortalities likely attributable to females that breed on St. George Island and Sea Lion Rock, based on the equilibrium population estimates for St. Paul and St. George Islands, and Sea Lion Rock presented by Lander [5], with the result increased by 36% to account for female seals wounded or killed at sea but were not recovered [67].

*2.5. Bayesian Estimates of St. Paul Fur Seal Survival as a Function of Pollock Catch*

We use the deterministic Leslie matrix, *A*, described above, to develop informative prior distributions for fecundity and survival for a model of random pup recruitment. We call the first row of *A* the transpose of the vector *f*. The observed data are $N_{0,y}$, that is, the observed number of female pups born in year *y*. *S* is a vector of time invariant random probabilities of age $i-1$ females surviving to age *i*, for *i* = 2, 3, . . . 24+.

We assume the random fecundity parameters $\varphi_i$ follows a beta distribution with parameters $10f_i$ and $10(1-f_i)$:

$$\varphi_i \sim \text{beta}(10\,f_i, 10\,(1-f_i)), \text{ for } i = 0, 2, \dots 23.$$

Note that the prior probability for fecundity is moderately concentrated around the values given by Lander [5]. We assume that the precision of the recruitment function follows a gamma distribution:

$$\tau \sim \text{gamma}(10^{-2}, 10^{-2}).$$

Similarly, we assume the random population survival parameters $\xi_i$ follow a beta distribution:

$$\xi_i \sim \text{beta}(30\,s_i, 30\,(1-s_i)), \text{ for } i=0, 1, 2, \dots 24+,$$

This prior probability is heavily concentrated around the estimates derived from Lander [5]. We then assume that the unobserved parameters that describe the age-class abundances and distribution are given by

$$N_{i,y} = \xi_i N_{i-1,\,y-1} \text{ for } i = 1, 2, \dots 24+. \tag{6}$$

If needed, the likelihood function can then be found from the sampling distribution for pup births, which we assume is normally distributed for each time step *y*:

$$N_{0,y} \sim \mathbf{N}(\boldsymbol{\varphi'N_{y-1}}, \tau^{-1}). \tag{7}$$

We assume an equilibrium age class distribution for the 1955 age class, scaled so that the age-0 abundance matches the estimated female pup births in 1955. We subtract age-specific females harvested from 1956 through 1974 [6] by year, from the annual age-class distributions (Equation (7)) as the random process advances through time. Additionally, starting in 1990, pup surveys have been conducted every other year. Missing values could have been treated as unknown parameters, but as a simplifying step we simply impute missing values by averaging the observation before and after the missing value. This simplifying step seemed justified by the moderate auto-correlation ($r = 0.710$) in the pup recruitment series beginning in 1955.

The probability of transitioning from age 0 to age 1 (i.e., pup survival) was our parameter of interest. Marginal posterior distributions for pup survival of female pups were calculated using Markov chain-Monte Carlo sampling, as implemented in JAGS running with R version 3. All simulations of these parameters used 150,000 or more

simulation draws, and we used at least 100,000 simulation draws to inspect the marginal posterior distributions of other parameters and quantities. To evaluate the evidence that the EBS pollock catch affected pup survival, a posterior distribution was estimated for a case where there was a single random pup survival parameter for all years, and then a second case with a separate parameter for: (1) years from 1955 to 1963, before the advent of the pollock fishery; (2) years of low catch (<750,000 t); (3) years of moderate catch (750,000 t < catch < 1,050,000 t); and (4) years of high catch (>1,050,000 t). We chose these catch limits so that the three catch levels have nearly equal numbers of years (i.e., low, 17; medium, 19; and high, 19). Models were compared with the deviance information criterion (DIC) [68] using JAGS. We assumed that a model with a DIC value seven or more units smaller to be superior.

## 3. Results

### 3.1. Effects of Fur Seal Recruitment Perturbations

Reducing adult survivals by 10% or 50% for 5 years immediately reduces the number of subsequent pup births, but reducing pup survival has little or no effect on pup births until 5 or more years later (Figure 3). In the cases of a reduction in the survival of adults, when survival returns to a base level, pup births increase somewhat because of the recruitment of juveniles that had not yet reached adulthood after survival was lowered. The complete loss of a pup year class reduces pup births over several years beginning 5 years after incidence, after which pup births stabilize at a lower level. Reduction in pup survival by 50% for five years reduces pup births over a period of five years beginning about 5 years after incidence, reaching a maximum reduction in pup births after 10 years, followed by a slight recovery to a lower equilibrium. In all four survival reduction scenarios, the lower equilibrium level of pup births results from the absence of mechanisms for population recovery in our Leslie matrix model, so this model cannot replace population losses from the modeled survival reductions.

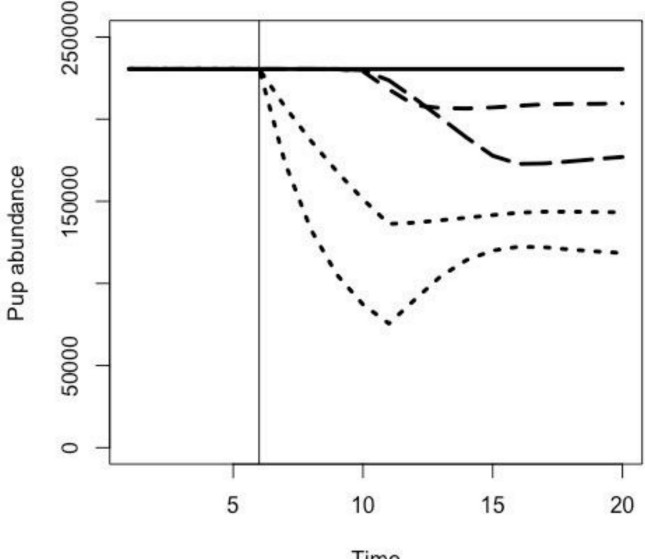

**Figure 3.** Hypothetical pup abundance projections over a 20-year period under different manipulations of Lander's Leslie matrix [5] for northern fur seals. The horizonal line at the top shows the projection of annual pup births with no manipulation. For all other curves, there was a change in year 6 (shown by vertical line). The two dotted lines show projections if adult seal survival is reduced by 10% and by 50% for five years. The curve with short dashes shows the effect of the complete loss of one year class of pups. The curve with long dashes show the projection if the first-year survival of pups is reduced by 50% for five years. At the end of these perturbations survivals are modeled to return to equilibrium values. Note that reduced adult survival has an immediate effect on pup births, whereas reduced pup survival has a lagged effect on pup births.

Cross-correlations between pollock catches and fur seal pup births are consistently negative, and greatest when pup births lag pollock catches by 10 to 12 years (Figure 4A). Weaker cross-correlations are also evident near $0 \pm 1$ year lag, and when pollock catches lag pup births by 11 y. The stronger correlation when pup births lag pollock catches at 11 years largely results from the period from 1964 through 1967, when pollock catches estimated by [23] were relatively low and the fur seal population (and hence pup births) was relatively high (Figure 4). Close examination of Figure 4B shows a series of seemingly serial correlated clusters that are above the smooth line early in the series (e.g., 68–70, indicating pollock catches from 1968 to 1970) and that move far below the smooth line later in the series (e.g., 0–7, indicating pollock catches from 2000 to 2007). Starting in 1988, as pollock catches approached or exceeded 1,000,000 t, the linear relationship between pollock catch and pup births that drove the elevated correlation appears to largely disappear, as indicated where the LOWESS smoothing curve in Figure 4B becomes more horizontal.

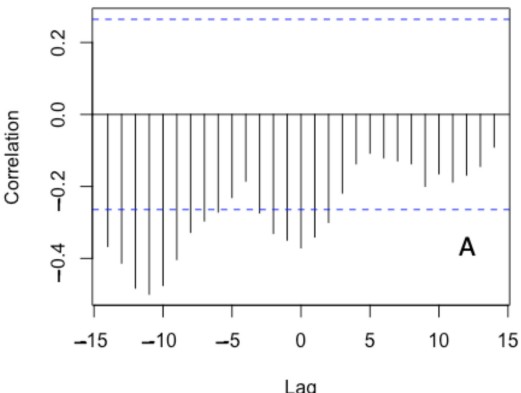
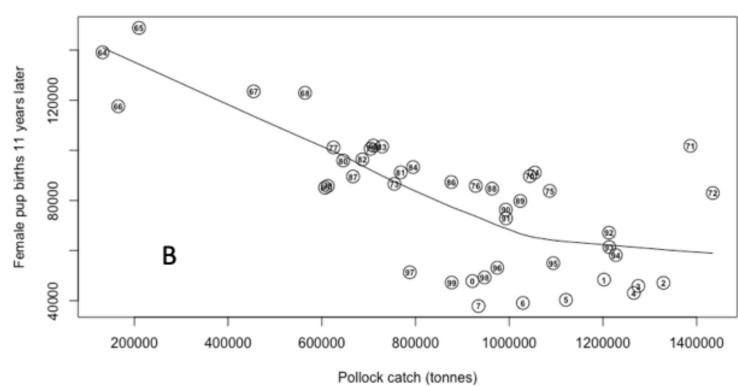

**Figure 4.** (**A**) Cross-correlations between catches of pollock at different lags within our Pribilof fur seal foraging area, and female northern fur seal pup births on St. Paul Island from 1964 through 2018. Pollock catch leads pup births and is inversely related to female pup births, with a peak correlation at −11 years. (**B**) St. Paul fur seal pup births as a function of pollock harvest 11 years earlier. The line in B is a LOWESS smoothing curve. Much of the appearance of a negative correlation comes from the three data points from a period of very low harvests of pollock in the 1960s, which are clustered in the upper left corner of the graph. The two-digit numbers used for plotting represent the years of the pollock catch in the 1990s. The one-digit codes indicate the year in the 2000s.

To understand the possible mechanism, both the change in pollock catch and the secular change in the fur seal population size in time must be considered. The change in time is linked to the decline in the breeding population size, which in turn leads to further declines in fur seal pup births. Near or above pollock catches of 1,000,000 t, a brief period of high catches (e.g., 1970 to 1972) appears to be associated with a very different trajectory in pup births than the later sustained high pollock catches after the early 1990s, which are associated with a steeper relative decline in subsequent pup births (Figure 2A)

Cross-correlations between pollock biomass and fur seal pup births are also consistently negative and are greatest when the lag between pollock catches and pup births is $0 \pm 3$ years (Figure 5). A much weaker cross-correlation occurs when pup births lag pollock biomass by about 11 years. The stronger correlation at year 0 also largely results from the period from 1964–1967, when estimated pollock biomass was relatively low and the fur seal population (and hence pup births) was still relatively high, estimates from the 1980s and 1990s when estimated biomass was high but pup births had fallen to uniformly low levels, and periods of high biomass in the 2000s and 2010s when pup births were consistently low (Figure 5).

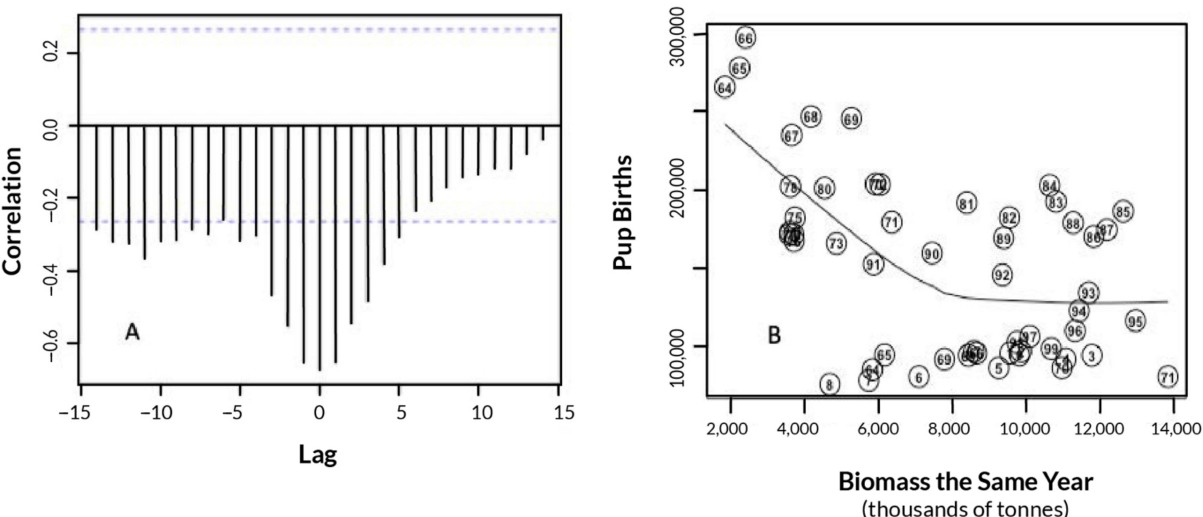

**Figure 5.** (**A**) Cross correlation between estimated eastern Bering Sea biomass at different lags and female northern fur seal pup births on St. Paul Island from 1964 through 2018. The peak in correlation is negative at a lag of zero, with a much smaller peak at a lag of −11, where biomass is slightly negatively correlated with female pup births in the future. (**B**) St. Paul fur seal pup births as a function of pollock biomass in the same year. The line in B is a LOWESS smoothing curve. The two-digit numbers used for plotting represent the years of the pollock catch in the 1990s. The one-digit codes indicate the years in the 2000s.

### 3.2. Pup Survival Perturbation Models

Model 1: Our model relating reduced pup survival through a logistic function of pollock catch from within our Pribilof fur seal foraging area closely tracks the observed pup production (Figure 2A). Residuals of the fit of Model 1 to production estimates on St. Paul Island showed little evidence of patterns or trends, when expressed as a proportion of the model estimates (Figure S1, Supplemental Information). The estimated values of $C_0$ and $k$ are 1,230,000 t and $5.69 \times 10^{-6}$ t$^{-1}$, respectively. The logistic function used to model the relationship between pollock catch and pup survival (Equation (3)) estimates pollock catches that reduce month-5 pup survival to 90% and 10% of the equilibrium value of 0.592 at 839,000 t and 1,610,000 t, respectively (Figure 6).

Model 2: Compared with catch, our model relating pup survival with a type II functional response to total age 3+ pollock biomass tracks the pup production time series less well (Figure 2A). Residuals of the fit of Model 2 to production estimates on St. Paul Island showed clear trends of persistent deviations, with modeled estimates consistently lower than field estimates from 1971 through 2000, and consistently above from 2003 through 2018 (Figure S2, Supplemental Information). The estimated values of $a$ and $h$ are 19.7 and 2.09, respectively. Corresponding values of $a$ and $h$ for total age 1+ biomass and total ages 1–3 biomass were 11.2 and 2.04, and 1.96 and 1.56, respectively. Plots of predicted female fur seal pup births based on the age 1+ biomass and the ages 1–3 biomass were nearly identical with the plot based on the age 3+ biomass as presented in Figure 2A, and the residual plots for all three curves were also nearly identical. The asymptotic values of the functional response component of Equation (4), $1/h$, for total age 3+ biomass, total age 1+ biomass, and total ages 1–3 biomass were 0.478, 0.490, and 0.641, respectively. The total ages 1–3 biomass associated with the equilibrium first year fur seal survival is equivalent to 13,500,000 t, which is more than 40% above the highest estimates of total ages 1–3 biomass since the fishery began in 1964.

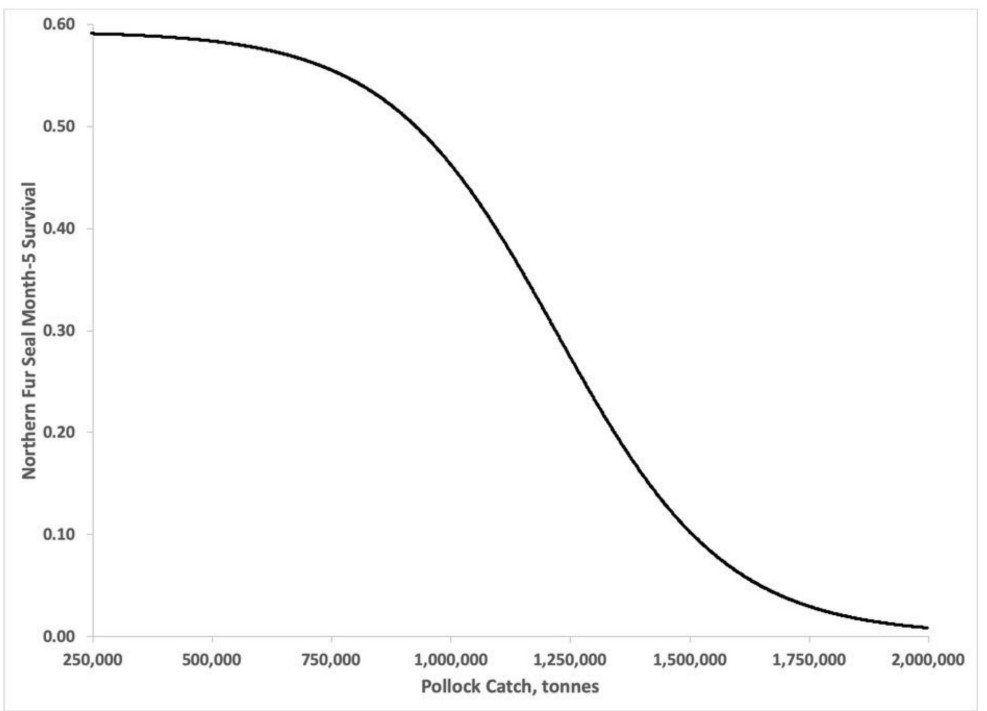

**Figure 6.** Logistic function modeling the relationship between pollock catch (t) within our Pribilof fur seal foraging area and month-5 survival of fur seal pups (Equation (3)).

Results for the combined effects of pollock catch from within our Pribilof fur seal foraging area and any of the three total pollock biomass scenarios (Model 3, based on total age–3+ biomass, total age–1+ biomass, or total ages 1–3 biomass) are nearly identical with the effects of pollock catch (Model 1) alone ($\hat{\pi}$ = 0.978).

### 3.3. Bayesian Estimates of St. Paul Fur Seal Survival as a Function of Pollock Catch

The Bayesian posterior distributions of the first-year survival of St. Paul fur seal pups when a single survival term $s_0$ is estimated throughout the 1955 through 2018 record has a mean value of 0.58 (S.D. = 0.066). The marginal distributions of the fecundity parameters generally tracked those of Lander, although they tended to be slightly smaller (e.g., posterior median of 0.38 vs. 0.42 for 9-year-old females from Lander or posterior median of 0.42 vs. 0.44 for 11-year-old females from Lander [5]). Similarly, the marginal posterior distributions for the survivals tended to be slightly smaller than Lander's values [5] (e.g., a posterior mean of 0.92 vs. 0.95 from Lander for the probability of an age-9 female surviving). When the model was expanded to allow four different parameters based on the level of pollock catch the DIC values decreased by a value of 23. Therefore, the four-parameter model is preferred by our criterion.

When the model was allowed to estimate four different survival values, the posterior distribution for the first-year survival estimates has a mean of 0.62 (S.D. = 0.055) for years before the pollock fishery began (i.e., 1955 to 1963), 0.75 (S.D. = 0.055) for low pollock-catch years, 0.62 (S.D. = 0.064) for medium pollock-catch years, and 0.25 (S.D. = 0.052) for high pollock-catch years (Figure 7). Here, too, the marginal posterior distributions of adult survival and fecundity had means that closely tracked Lander's values, as expected.

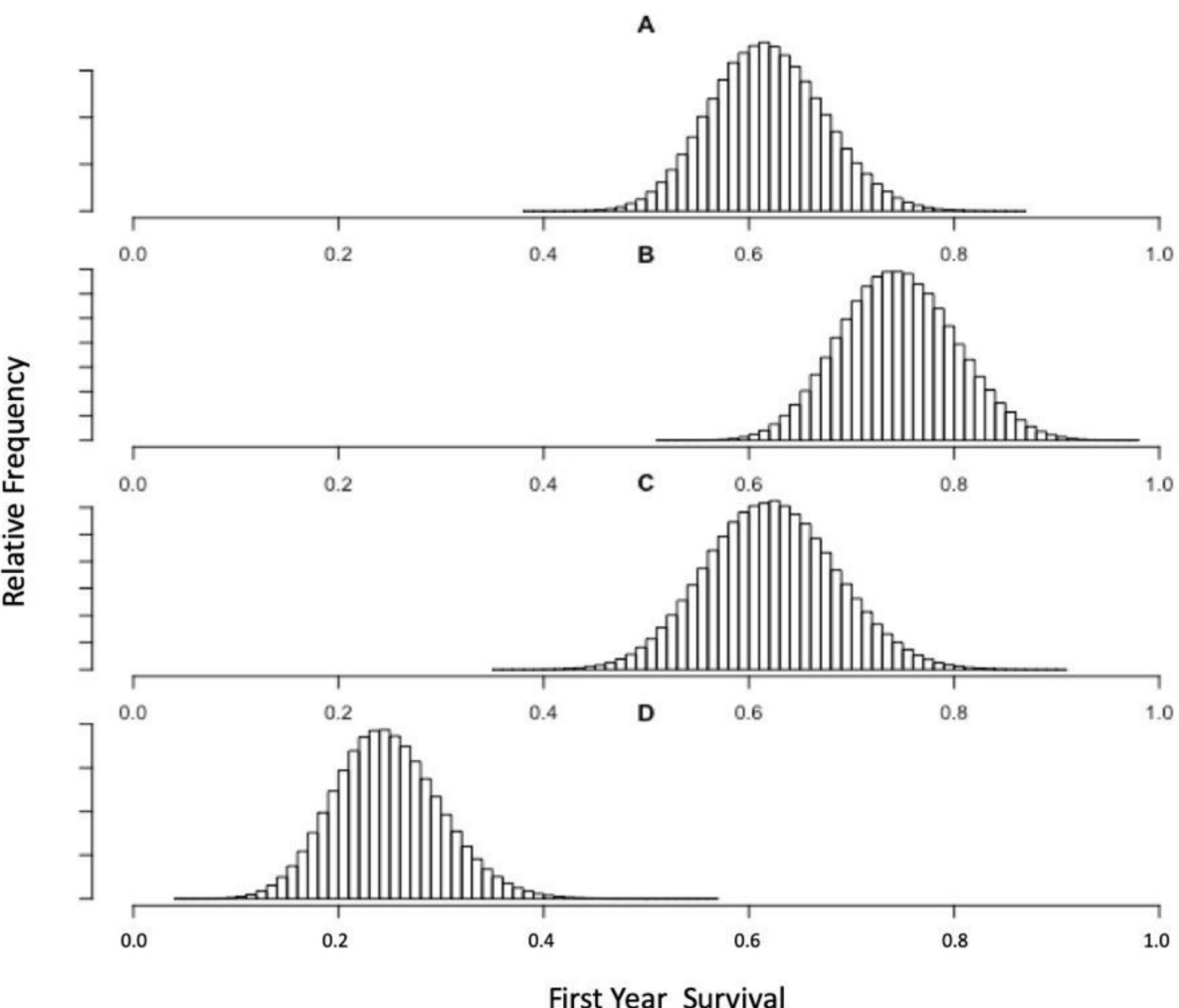

**Figure 7.** Posterior distributions for the probability of first-year survival for St. Paul fur seal pups following four categories of eastern Bering Sea pollock catch in the same year. (**A**) Distribution for the years 1955 to 1963, prior to the rapid increase in pollock catches. (**B**) Distribution for years classified as periods of low catch (<750,000 t). (**C**) Distribution for years of moderate catch (750,000 t < catch < 1,050,000 t). (**D**) Distribution for years classified as periods of high catch (>1,050,000 t).

## 4. Discussion

### 4.1. Comparison of Pollock Catch and Biomass Models

The evidence we report is supportive of our operating hypotheses: (1) the annual magnitude of northern fur seal births is determined by both the abundance of breeding females and the survival of fur seal pups; and (2) first-year survival has been affected by the magnitude of the pollock catch in the EBS. If the pollock fishery adversely affects pup survival, then time must elapse between the reduction in survival and detection of this reduction, which is when the missing females would have first started to produce their own pups, as pup births are the principal population metric. Additionally, because the fishery catch level is auto-correlated, if the pollock fishery reduces pup survival, this reduction should be somewhat persistent.

All three of our approaches for evaluating the relationship between pollock catch and lagged declines in St. Paul fur seal pups imply substantial correlation. The elevated inverse cross-correlations between pollock catches and pup numbers 10 to 12 years later (Figure 4A) are consistent with lags expected from reduced survival during their first year of life, and consequent reductions in pup births that result from the lower number of female seals that survive to adulthood, especially when reduced pup survivals span several consecutive years (Figure 3). Spurious correlations that arise from the considerable auto-correlation in

both the pollock catch and the pup production time series cannot be dismissed. We also recognize that the inverse correlation between higher pollock catches and reduced pup births after a lag of 10 to 12 years results in considerable part from leverage associated with the earliest years of both the pollock catch and pup production time series (when catches were low and pup production was highest). Nonetheless, the inverse correlation remains suggestive of an inverse relationship between high pollock catches and reduced pup survival in the same year, which ultimately motivated our further investigations.

Our pup survival perturbation model (i.e., Model 1) provides stronger support for an inverse relationship between pollock catches and pup survivals during their first year. Our Leslie matrix model of population trajectory is a simplified approximation, lacking many features of real populations, including density dependence and random process error. Yet, when this deterministic model is modified so as to only allow the pup survival to vary as a function of pollock catch and to account for direct mortalities due to female seal harvest earlier in the series, the model is able to reproduce the main features of the pup birth data series from 1950 to 2018. Specifically, this model tracks the four main declines during 1955 to 1963, 1979 to 1981, 2000 to 2010, and 2014 to 2018. Furthermore, considering the high catches of pollock from 2011 to 2018, exceeding 1,046,000 t annually within our fur seal foraging area, this model predicts another period of seal pup birth decline beginning around 2018 and continuing for another six years or so. Notably, adding additional complexity to this model did not improve its ability to track the pup birth series, and a similar model using three metrics of pollock biomass (i.e., total age 3+ biomass, total age 1+ biomass, and total ages 1–3 biomass), each evaluated separately, was not able to reproduce the series of pup births nearly as well. At its simplest level, the Leslie matrix approach is based on the assumptions that fecundity and survival for all ages older than pups are stable over time, and that these parameters are essentially known. If these assumptions are met, then recruitment is predictable and is controlled by the size of the breeding population. With this simple, modified algebraic model, both the observed declining size of the breeding population and occasional lowering of pup survival seem to explain the population trajectory surprisingly well.

The posterior distributions from our Bayesian estimates of first year survival probabilities provides yet another way to organize the evidence for an inverse relationship between pollock catches and pup survivals during their first year. The highest estimated survival probabilities (i.e., 0.75), associated with years when pollock catches were lowest, imply the population had an underlying tendency to expand. This underlying tendency was largely negated during years of medium pollock catches, when first-year survivals were similar to those during the fur seal reduction program from 1956 through 1974. During years of high pollock catches in our fur seal foraging area (>1.05 million t), however, first year survivals were clearly depressed in comparison with the other pollock catch categories.

In contrast, the results of our analyses to evaluate the strength of relationships between pollock biomass indicate that pollock biomass had little effect on fur seal pup production during the period of declining fur seal abundance we evaluated. The inverse relationship found by our cross-correlation analysis (Figure 5), implying that greater pollock biomass somehow reduces pup survivals both before and after a given year of high pollock biomass, suggests an artifact perhaps resulting from the relatively high auto-correlation in the pollock biomass and pup birth time series. In addition, there is considerable uncertainty regarding the size of the pollock population at the beginning of the fishery (late 1960s through at least the mid-1970s). Stock assessments of EBS pollock conducted prior to the late 1990s (e.g., [69]) estimated that biomass peaked in the early 1970s at between 11 and 13 million t as a result of the recruitment of strong year-classes in 1965–1969. This is approximately double the early-1970s biomass peak estimated by [23], but the only new data included in the 2019 assessment are Japanese fishery catches per unit effort (CPUE), which are unreliable estimators of stock biomass for a highly aggregated species such as pollock [70]. The early fishery (prior to the mid-1980s) caught many more small, young (age 1–3) pollock than it has over the last 30 years [23]. Changes in fishing technology

and targeting practices [71,72] have contributed importantly to temporal differences in fishery size-selectivity. Nonetheless, large catches of small fish have led some to conclude that the pollock population in the 1960s and early 1970s was 'young' and small [73–75], a contention disputed [76] given the pattern of recruitment to the population and catches of age 4+ pollock throughout the time series [23]. If pollock biomasses in the early part of the time series (1964–1973) were substantially greater than estimated by Ianelli et al. [23] and more similar to those estimated by Bakkala et al. [69], then many of the 'low biomass-high pup production' points in Figure 5 become 'high biomass-high pup production' points, further weakening the apparent inverse correlation. Moreover, the results of the pup survival perturbation model based on pollock biomass (i.e., Model 2) seem unrealistic, given the values estimated for the *h* parameter of the type II functional response equation. These values imply that the fur seal population would always decline regardless of the magnitude of the total age-1+ or age-3+ pollock biomass, and would decline with any of the estimates of total biomass of ages 1–3 since the fishery began. Overall, our results suggest that fur seal pup survival is relatively insensitive to pollock biomass when fur seal numbers are relatively low, but is sensitive to factors associated with pollock catches.

### 4.2. Pollock Catch and Regime Shift Effects on Fur Seal Births

We suspect that the inverse relationship we infer between pollock catches in the EBS and Pribilof northern fur seal pup survival arises from competition between the pollock fishery and lactating female fur seals for high-density aggregations of pollock. The distribution of pollock in the EBS is not homogenous, but includes high-density aggregations where zooplankton and nekton species that are consumed by pollock are concentrated by oceanographic convergences and retention areas [77]. The pollock fishery targets these aggregations, and their remnants after being fished are both depleted and fragmented into smaller dispersed schools [28,78]. Lactating northern fur seals also target dense aggregations, because successful location of them close to the Pribilof Islands increases the rate that females can deliver milk to their pups [28].

McHuron et al. [79] showed that adult female fur seals breeding on the Pribilof Islands may be expending the maximum amount of energy possible to obtain food during foraging trips. They may have reached this metabolic ceiling at least by the mid-1990s, and the possibility that this limit was reached as early as the 1970s cannot be dismissed. As a consequence, food limitation may be contributing to reductions in pup mass gains prior to weaning because less energy is available to support lactation. Longer foraging trips result in fewer nursing opportunities, which yields less energy transfer to pups [26,51,80]. As a consequence, pups are lighter at weaning in the fall, which in turn lowers first-year survival [81]. Scheffer [11] postulated that Pribilof fur seals had reached their natural carrying capacity by the late 1940s-early 1950s and that the population stopped growing because of density-dependent food limitation: " . . . as the herd increased, the competition for food among the hundreds of thousands of nursing mothers grew keener; the average mother had to swim farther from the islands to find fish and squid; she returned less frequently to feed her baby; the pup put on less weight; more pups died in their first year; and the herd decreased." The competition between lactating female fur seals and the fishery for pollock that may be currently reducing pup growth and survival [26,51,79] has occurred when Pribilof fur seal numbers are less than 25% of those in the late 1940s, and are more similar to those observed shortly after the cessation of pelagic sealing and the signing of the North Pacific Fur Seal Convention in 1911 [1–3]. Thus, Pribilof fur seals are responding as though they have exceeded the carrying capacity of their foraging habitat, but at progressively-declining population sizes. This suggests that there has been a dramatic reduction in the ability of the EBS to support northern fur seal breeding requirements, and that this change occurred between the early 1970s, after the majority of the effects of the culling of adult females had passed through the population [6], and the mid-1990s, when McHuron et al. [79] first demonstrated that adult females were foraging at their metabolic ceiling.

The EBS pollock fishery in effect introduced a major new predator into the EBS ecosystem that directly competes with the Pribilof northern fur seal herd beginning in the mid-1960s (Figure 2B). Catches were low in our fur seal foraging area through 1969, so there should have been little effect of catch on pup births through the mid-1970s, which is what is observed. In fact, there was a slight increase in pup births in 1973–1976, perhaps a density-dependent response in the now smaller fur seal population and evidence that the effects of the adult female harvest programs had ceased. However, beginning in 1970, foraging area catches were high for 6 of the next 7 years (Figure 2). As we have shown, the effects of high pollock catches on fur seal population dynamics do not manifest themselves for at least 5 years, perhaps peaking at 11 years. The early 1970s high-catch period matches well with the decline in pup births on St. Paul observed in the late 1970s-early 1980s. Given that pollock is the primary prey item of Pribilof fur seals during the breeding season and that there is a large spatial overlap between foraging areas used by Pribilof fur seals and the fishery (Figure 2A), competition between the pollock fishery and Pribilof fur seals could occur at least at two spatial-temporal scales: long-term (decadal) across the entire EBS shelf in the US, and short-term (annual) in areas used by both the fishery and fur seals.

Fishery-induced, decadal scale, and ecosystem-wide decreases in average annual pollock biomass are likely to have occurred at least to some degree, but may have been mitigated by the strategy used to manage groundfish in the EBS since passage of the US Fisheries Conservation Management Act (FCMA) in 1977. The early fishery (1964–1972) was prosecuted entirely by foreign fisheries (principally from Japan and the former Soviet Union) and was essentially unregulated; catches were relatively low through 1969 but increased dramatically in 1970–1972 [59,60] (Figure 2B). Concerns regarding the collapse of heavily-fished stocks in the eastern North Pacific prompted implementation of pollock catch quotas in 1973 by the International North Pacific Fisheries Commission [69]. Passage of the FCMA, which was done in part to encourage the development of US domestic fisheries, required the implementation of Fishery Management Plans (FMPs) to obtain the maximum sustainable yield (MSY) from each stock while avoiding over-fishing. The initial FMP for Bering Sea and Aleutian Island (BSAI) groundfish was finalized in 1981 (and updated in 2020 [82]), and established an optimum yield (OY) for the entire complex (including pollock) in the range of 1.4–2 million t, or about 85% of the sum of the single-species MSY values available at that time (82). The sum of the catch quotas for all species of groundfish could be no greater than the upper limit of the OY range (2 million t). To allow for the prosecution of many other high volume groundfish fisheries, annual pollock catch quotas implemented since 1981 have been constrained by the 2 million t OY cap and have been lower than what they could have been without it [23,60]. Since 1981, annual exploitation rates have been mostly below 20% and often approached 10%, and pollock catch magnitudes since 1981 have been somewhat decoupled from changes in the underlying biomass that supports the fishery. This, we believe, may have contributed to the difference we observed in the magnitude of the effect of pollock catch vs. pollock biomass on fur seal pup production. Based on our analyses, changes in overall biomass of EBS pollock (regardless of the cause) explained far less of the variation in fur seal pup production than pollock catch. This leads us to conclude that the driver of the fishery effect is not the rate at which pollock is removed from the EBS or how much remains after the fishery, but the magnitude of the catch or something directly related to the catch amount.

In contrast, fishery effects occurring annually within our fur seal foraging area (Figure 1D) include disturbance of fur seals by fishing vessels, reduced density of pollock schools, reduction in the number of high-density schools, and creation of a more diffuse and less predictable prey field [83–85]. Indirect fishery effects tend to decrease fur seal foraging efficiency by making them expend more energy to find sufficient food, consistent with the results of McHuron et al. [79]. Otariid pinnipeds select the densest fish schools and forage repeatedly in areas where they have previously encountered dense aggregations of prey in order to optimize their foraging efficiency [28,86–92]. These areas are selected over other areas that may have more biomass but not as many high-density

aggregations [28,86,88,89]. A decrease in prey density as small as 15% was large enough to have demographic consequences for Steller sea lions [93]. Analyses of hydroacoustic data collected continuously during pollock fishing revealed that repeated trawling through aggregations in the southeastern Bering Sea reduced the number of high-density schools of both adults (targeted by the fishery) and juvenile pollock, created a more dispersed pollock distribution, and coincided with a significant decline in the catch efficiency of the fishery itself over a month-long period [78]. Some of these effects (e.g., reduced number of high-density schools, creation of a more dispersed prey field) are likely to scale at least linearly with catch magnitude and be relatively independent of the biomass remaining in the area after the fishery. Our results regarding the relationship between EBS pollock catch and fur seal pup production are consistent with and help explain those of McHuron et al. [79], as well as Scheffer's [11] carrying capacity model of fur seal response, except that, in the case of Scheffer's model, the pollock fishery is substituted for many hundreds of thousands of fur seals.

Another change in the North Pacific ecosystem that has been hypothesized to have reduced the carrying capacity for Pribilof northern fur seals between the early 1970s and the mid-1990s was the 'regime shift' of 1976–77 [44,73,75]. The proposed mechanism can be summarized as follows: (1) warming of the waters in the North Pacific Ocean and EBS in 1976–77 was associated with a step-change in oceanographic and atmospheric indices suggesting they were one phase of a natural cycle between warm and cool states [94,95]; (2) warm waters led to a change in fish community structure due to favorable conditions for recruitment of prey species with low energy density (e.g., gadids such as pollock and Pacific cod) and unfavorable conditions for high energy prey (e.g., 'forage' fishes such as eulachon (*Thaleichthys pacificus*), capelin and Pacific herring) [95–102]; and (3) northern fur seals could not obtain enough energy eating predominantly low energy prey which led to nutritional-stress and population declines [44,73,75] (also see review of this hypothesis in [74] with respect to the decline of Steller sea lions). Adult females would be the most likely sex-age group affected by this type of regime-shift-induced nutritional stress in the EBS, and the effects would manifest in ways similar to those described for pollock catch: reduced pup survival due to low weights at weaning which would lead to reduced pup births 5+ years later. However, pup numbers declined the same year as the regime shift and for the next 3 years (1977 through 1980). The regime shift has been linked to the recruitment of two of the largest pollock year-classes on record (spawned in 1978 and 1982) which led to high pollock biomass that peaked in the mid-1980s [23]. Additionally, instead of declining as pollock biomass increased, fur seal pup births were relatively stable through the 1980s and well into the 1990s, a pattern that is inconsistent with the regime-shift fish-community reorganization hypothesis. Fritz and Hinckley [76] also reviewed the evidence for the regime shift hypothesis in the EBS with respect to both Steller sea lion and northern fur seal declines and found no evidence to support the necessary component that gadid recruitment is low during cool phases and high during warm phases. Nor did they find evidence to support the contention that a gadid-rich diet should necessarily lead to poor recruitment in Pribilof fur seal populations, since there is ample evidence that they consumed large quantities of pollock as their numbers increased steadily in the first half of the 20th century in both warm and cold regimes (see review by Sterling [51]). Finally, if regime-mediated changes in pollock recruitment (rather than fishery catch) is the chief driver of fur seal population dynamics, then we should see a stronger correlation between pollock biomass and fur seal births than between catch and births, but we do not.

*4.3. Fur Seal Size Selectivity of Consumed Pollock*

Swartzman and Haar [65] reviewed data available through about 1980 relevant to the unexplained decline of fur seals after the adult female harvest ceased, and the possible role of the then relatively new pollock fishery. One of their hypotheses was that fishing pressure on cannibalistic adult pollock led to an increase in the abundance of juvenile pollock. This in turn led them to conclude that after the introduction of the fishery, Pribilof fur seals

increased the proportion of pollock in their diets because, they argued, (1) fur seals prefer juvenile to adult pollock, (2) juvenile pollock were concentrated (hence readily available) near the Pribilof Islands, and (3) fur seals did not have to forage as far away from the islands in search of Pacific herring and capelin. With the benefit of a considerable amount of data collected since Swartzman and Haar published their paper [65], we now know why their conclusion that increased fishing pressure led to greater pollock consumption was incorrect. First, catch-at-age data from the early (1964–1974) fishery reveals that young, not old, fish were targeted. This does not mean that there were few old fish in the population, only that the fishery predominantly targeted and caught young fish. Therefore, it appears unlikely that an increased abundance of juveniles was the result of removals of cannibalistic adults, but was instead the result of environmental factors that favored recruitment in the late 1960s (1965–1969) [69]. Unfortunately, there are no reliable independent survey data available from this time period to describe the true pollock age structure during the first decade of the fishery. Second, it is now known that fur seals consume both juvenile and adult pollock, and that interannual variability in the sizes or ages consumed depends on the recent recruitment history of pollock [47,51–54]. If recent recruitment was strong (i.e., abundant age 0, 1 or 2 years old), then fur seals will eat primarily juvenile pollock in the upper water column. However, if recent recruitment was poor, then fur seals will eat older, adult fish found near the bottom, necessitating deeper dives. Third, conclusions regarding a shift in Pribilof fur seal diets from Pacific herring and capelin (high-energy density prey) to pollock (low-energy density prey) at about this time due to the 1976–1977 climate regime shift ignore not only critical metadata associated with diet collections of the 1960s, 1970s and 1980s, particularly regarding the locations where pelagic fur seal stomach samples were collected and the shift to analysis of scats collected on land to analyze diet (see review in [51]), but also the well-documented and long history of fur seal consumption of pollock, including adults [14,15,25,33,39,41,45,46,103].

### *4.4. Fur Seal Population Response to Competition*

Pribilof fur seals began colonizing Bogoslof Island in 1980, and numbers have steadily increased since [4]. Numbers of fur seals colonizing Bogoslof Island remained modest through the mid-1990s, increasing from 7 in 1980 to ~1,400 in the mid-1990s [104,105]. During this period, female northern fur seal pup production on St. Paul Island was relatively stable, fluctuating around 100,000 per year (Figure 2). After 1995, numbers of fur seals colonizing Bogoslof Island increased to several thousands annually, with more than 5000 in 1997 and reaching nearly 28,000 by 2015 [4]. Seals colonizing the island are able to exploit nearby, lightly or unfished assemblages of Atka mackerel, capelin, bathylagid smelts and squid [103]. Competition for high density pollock aggregations between northern fur seals and the fishery may have contributed to the displacement and range expansion of breeding seals from the Pribilofs to Bogoslof Island [106,107], an unusual response by a declining population. Breeding-range expansion by declining Pribilof fur seals is similar to that of declining western Steller sea lions as they established new rookeries in northern southeast Alaska, also in response to resource limitations in their original range [108].

### *4.5. Restorative Capacity of the Pribilof Northern Fur Seal Herd*

Our Bayesian estimates of first-year survival rates for all but the highest pollock catch category suggest that the Pribilof northern fur seal herd retains an intrinsic capacity to increase. These estimates, ranging from 0.62 to 0.75 (Figure 7), are substantially greater than the equilibrium survival rate of 0.50. We understand that these survivals estimates are not strictly comparable to Lander's Leslie matrix values [5], as the posterior values for the fecundity and survival that are comparable to our pup survival estimates are slightly different. We note that the pup survival values we estimated for years when the pollock catches were the lowest (i.e., 0.75), even after the pollock fishery expanded, should be high enough for the seal population to eventually expand if this survival level could consistently be achieved. Even the level we estimated for years of moderate harvest (0.62) might allow

a stable if not slightly increasing population. Taking the eigenvector of Lander's Leslie matrix, with the pup survival value changed to 0.62, we get $\lambda$= 1.0195, which predicts an approximate 2% population increase per year. Using the same logic, Lander's Leslie matrix with pup survival reduced to 0.25 should lead to an approximate 6% decline per year. While these cannot be taken as precise forecasts of the population response, they do suggest that the Pribilof northern fur seal herd retains its inherent capacity to rebuild, as was reflected by the herd's rapid response to conservation measures after the previous two steep declines in herd size during the 1820s–1830s and the 1880s–1900s.

**Supplementary Materials:** The following are available online at https://www.mdpi.com/article/10.3390/jmse9090975/s1, Figure S1: Residuals of data fit to the pollock-catch-modified Leslie matrix model of northern fur seal pup births, 1955–2018, Figure S2: Residuals of data fit to the pollock-biomass-modified Leslie matrix model of northern fur seal pup births, 1955–2018.

**Author Contributions:** Conceptualization, J.W.S., H.J.G., L.W.F. and J.J.W.; methodology, J.W.S. and H.J.G.; investigation, J.W.S., H.J.G., L.W.F. and J.J.W.; data curation, J.W.S., H.J.G., L.W.F. and J.J.W.; writing—original draft preparation, J.W.S., H.J.G., L.W.F. and J.J.W.; writing—review and editing, J.W.S., H.J.G., L.W.F. and J.J.W.; visualization, J.W.S.; supervision, J.W.S., H.J.G., L.W.F. and J.J.W.; project administration, J.W.S. All authors have read and agreed to the published version of this manuscript.

**Funding:** J.S., H.G. and L.F. were not compensated for their work on this manuscript. J.W.'s time was supported by Oceana.

**Institutional Review Board Statement:** Not Applicable.

**Informed Consent Statement:** Not Applicable.

**Data Availability Statement:** Data supporting our results that are not available in the cited references are available from the corresponding author (J.W.S.) on request.

**Acknowledgments:** We gratefully acknowledge B. Mecum for plotting and formatting map figures, and A. York for helpful comments on an earlier version of this manuscript.

**Conflicts of Interest:** The authors declare no conflict of interest.

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
