# Peer review of "First-Year Survival of Northern Fur Seals (Callorhinus ursinus) Can Be Explained by Pollock (Gadus chalcogrammus) Catches in the Eastern Bering Sea"

_jmse, doi:10.3390/jmse9090975_

Round 1

Reviewer 1 Report

Overall I found this paper well written and I think the topic will be of broad interest to both readers and folks outside the scientific community, like industry. I did have several some-what considerable concerns regarding the interpretation of the results and also a part of the analysis. I have outlined those concerns below, and have also attached a pdf with comments throughout highlighting these issues as well as other smaller comments. Hopefully the authors will find these comments useful in revising the manuscript.

  1. The authors base their main conclusion of the paper, that the fur seal decline is in part due to fishery removals of pollock, on a purely correlative relationship. This does not necessarily mean that the fishery is directly the cause of the reduction in pup production and I did not feel that the authors provided a good enough justification for this conclusion. There needs to be more discussion of what causes interannual variation in pollock catches, whether pollock catches could represent something biologically meaningful to fur seals (such as a good indicator of prey availability), and a better explanation provided as to why there was a good relationship with the catch data and not for the biomass data. This paper is bound to get a lot of attention from industry and I think it is therefore particularly important to be thorough in exploring the range of potential explanations that might explain the results and to be conscientious of not stepping outside the bounds of the data. 
  2. In their analysis, the authors analyze the fur seal decline across the entire time series, which assumes that the same mechanisms are causing the decline at all points. I don't necessarily think this is appropriate since there have been several distinct periods of decline, with the most recent and ongoing one starting in the late 1990s. This is particularly concerning given the main figure of the paper essentially shows that the relationship between catch and pup production completely breaks down for all years from ~ 1995 onwards. I think this issue needs to be explored with some additional analyses for each period of decline.

Author Response

Responses to Reviewer #1 Comments for Manuscript JMSE 1344625:

General Comments:

  1. The authors base their main conclusion of the paper, that the fur seal decline is in part due to fishery removals of pollock, on a purely correlative relationship. This does not necessarily mean that the fishery is directly the cause of the reduction in pup production and I did not feel that the authors provided a good enough justification for this conclusion. There needs to be more discussion of what causes interannual variation in pollock catches, whether pollock catches could represent something biologically meaningful to fur seals (such as a good indicator of prey availability), and a better explanation provided as to why there was a good relationship with the catch data and not for the biomass data. This paper is bound to get a lot of attention from industry and I think it is therefore particularly important to be thorough in exploring the range of potential explanations that might explain the results and to be conscientious of not stepping outside the bounds of the data. 
  2. In their analysis, the authors analyze the fur seal decline across the entire time series, which assumes that the same mechanisms are causing the decline at all points. I don't necessarily think this is appropriate since there have been several distinct periods of decline, with the most recent and ongoing one starting in the late 1990s. This is particularly concerning given the main figure of the paper essentially shows that the relationship between catch and pup production completely breaks down for all years from ~ 1995 onwards. I think this issue needs to be explored with some additional analyses for each period of decline.

Response:

We are sincerely grateful to this reviewer for the care, time and thought that he or she put into their review. It has prompted us to go over our manuscript in detail, add substantially to the text to address some of the concerns raised, as well as leading us to spot and correct a handful of minor errors.  We believe this has improved our manuscript considerably. 

That said, we believe this reviewer has misinterpreted some of the evidence we present, and has assumed we made stronger claims than we actually made.

The reviewer is correct in his or her opening sentence: Our main point is that we have found a purely correlative relationship between fur seal decline and the pollock. But what the reviewer does not seem to acknowledge is that the correlation is not just between pup births and the pollock fishery. Indeed, the strongest correlation is in a surprising place: a lagged relationship exactly where we would have expected to find it if the pollock fishery had affected first-year pup survival. It seems to us that this is a bit surprising, previously unacknowledged, and worthy of publication. Moreover, there is a plausible mechanism and an important public policy issue involved.

The reviewer seems to attribute much, much stronger conclusions to us than the actual conclusions we stated in the paper. Also, the reviewer seems to have missed the point of our operating hypothesis: “(1) the annual magnitude of norther pup seals is determined by both the abundance of breeding females and the survival of fur seal pups, and (2) first-year survival has been affected by the magnitude of the pollock catch…” If fur seal pup survival is reduced, this will not be observed for several years into the future. Moreover, if pup seal survival is reduced, this will, in turn, lead to fewer breeding females, which will continue to change the population dynamics even further into the future—just like the actual data appear to show. We showed, using three ways to organize the data, that the data is consistent with this hypothesis. We agree, this does not conclusively show that the fishery is directly the only cause of the decline—but we never claimed that. We do maintain that our analysis does add to the growing list of reasons to be more curious about the effect the pollock fishery has had on fur seals.

Finally, this reviewer appears to consider our cross-correlation analysis as the strongest evidence we present, when we thought we indicated that it is the weakest.  The main figure of the paper is Fig. 2A, not Fig. 4B, and Fig. 2A clearly shows that the relationship between pollock catch and subsequent pup births remains very close throughout the entire pollock catch record beginning in 1964 through 2018.  The results of the Bayesian analysis, which we performed to see if the credibility intervals for the effect of low, medium and high pollock catch magnitudes on pup survival were meaningfully distinct, clearly confirmed that they were.  These analyses (the deterministic models and the Bayesian analysis) are far more sensitive and realistic than the cross-correlation analysis.  We performed the cross-correlation analysis to see if there was any indication in the raw data that would justify more detailed analysis, which it did.  Nonetheless, this reviewer seems to think that because the pup births are consistently below the LOWESS smoothing function line after about 1995, it implies that "... the relationship between catch and pup production completely breaks down".  This is simply incorrect.  As we tried to point out at line 404 of the submitted manuscript, the main reason these points are so low after 1995 is because the population, and hence population fecundity, is only 27% – 63% of what it was in 1964.  So, the population cannot generate as many pups in 1995 as it could in 1964, and also the same reduction in month-5 survival probability generates a smaller reduction in the number of pups that survive in 1995 than it did in 1964.

All the same, this is a long, dense and rather complex paper spanning multiple disciplines, that we suspect most reviewers would find challenging at least in parts.  We strongly suspect it would have helped had we actually identified that the curves in Figs. 4B and 5B were LOWESS smoothing curves, and we regret not having done so, especially if the reviewer found these omissions misleading.  In recognition of all this we appreciate all the more the time this reviewer so clearly put into their review, and we sincerely hope this reviewer will find our efforts to address their concerns are satisfactory.

Specific Comments:

  1. 5 line 174: "...fur seal population estimates for St. Paul and St. George Islands are moderately correlated (r = 0.77, n = 23)."

Reviewer comment: This is probably not appropriate (I read this as implying that results for St. Paul would be applicable to St. George), particularly given the fact that the population has started to increase on St. George, whereas it is still declining on St. Paul. There are also some pretty distinct inter-island differences in diet, so I don't think that broad-scale generalizations can be made based on data from a single island.

Response:

This clause is simply a verifiable statement of fact. Either the correlation is 0.77 or it is not 0.77. This is not a statement of opinion, nor is it some kind of broad-scale generalization based on data from a single island. Indeed, this moderate correlation seems to show that there is some level of independence, which is exactly the statement of opinion that the reviewer is trying to make.  We included this clause to alert readers that to the fact that the general fur seal population trends of the two islands have had broadly similar downward trajectories over the last 5 decades.  Also, while the St. George population appears to have stabilized over the last 20 years, the evidence for a recent increase is at best tentative.

Regarding the inter-island differences in diet, we added text to the end of the paragraph that began on line 179 to address this point.

  1. 6 line 188: "Pollock is the dominant prey species in the EBS overall, especially around the Pribilof Islands and on the EBS shelf."

Reviewer comment: "It is worth mentioning here the inter- and intra-island differences in diet, which have been well documented"

Response:

We added text to the paragraph beginning on line 179 to address this comment, rearranged the flow within this paragraph to improve readability.

  1. 7 line 222: "Estimates of total (age 1+) pollock biomass in the EBS population beginning in 1964 are produced each year as part of the stock assessment process (Fig. 2, from [23]).

Reviewer comment: "I am assuming since they are from the SAFE reports that this is a total biomass for the EBS (ie. not foraging area specific)? This needs clarification (either way) here. It might also be worth it to look at the relationship with age 3+ pollock biomass to more closely align with what the fishery takes or the metric in numbers of fish. Also, Figure 2 from the citation is a figure of catch by sex not biomass"

Response:

We added text to make explicit that the total pollock biomass estimates referred to here is for the entire EBS population, including inside and outside our Pribilof fur seal foraging area, that the estimate is for ages 3+, and that the estimates are from Table 29 of reference 23, not Fig. 2.  We also added text here to explain how we estimated total pollock biomass for ages 1+ and for ages 1 – 3, which we use to provide the additional analyses to address this reviewer's comment on line 525 regarding use of more precise biomass metrics to evaluate possible relationships between pollock biomasses and fur seal recruitment.

Also, mention of Fig. 2 here was intended to refer the reader to our Figure 2, not the Figure 2 in the SAFE report cited.  We agree this was ambiguous and re-worded it to make it more clear.

  1. 7 line 227: "reduction of survival of all adults by 10% and separately by 50% for five consecutive years; reduction of pup survival by 50% for five consecutive years; and reduction of pup survival to zero for one year.

Reviewer comment: "What was the reasoning behind choosing these specific hypothetical scenarios?"

Response:

We added an introductory clause at the beginning of this paragraph to indicate our motivations for the hypothetical perturbations we presented.

  1. 9 line 299: " reduced availability of pollock that result from catch removals are not likely to increase the survival of fur seal pups;"

Reviewer comment: "You may need to address that this has actually been suggested in the past (i.e. removal of adult pollcock benefits fur seals since they are cannabilistic), although to my knowledge there is really no evidence either way on it."

Response:

We added two sentences here to address this suggestion, and a paragraph in a new subsection of the Discussion at line 687 that deals with this issue in more detail.

  1. 9 line 305: "...annual estimates of pollock catches"

Reviewer comment: "so both A and B season?"

Response:

We assumed that use of the word "annual" would imply the whole calendar year.  To reduce the possibility of ambiguity, we added "..., including all commercial fishing seasons within a calendar year,...",  here to explicitly state what we mean by "annual". 

  1. 9 line 307: "Model 2"

Reviewer comment: "Population models are not my forte so this may be a naiive question, but why wouldn't you model the catch and the biomass in exactly the same way? From Figure 2. it seems the purpose of this figure is to show that Model 1 (with catch) fits the observed pup counts better than Model 2 (with biomass) but these are not directly comparable if the models themselves don't have the same structures and assumptions"

Response:

The basic mathematical structures for these models are identical.  The only difference among them is the functional form relating the perturbation to month 5 survival.  Using a single function to model the effects of two quite different perturbations would introduce serious artifacts into the analysis.  The logistic function is well suited for modeling the effects of catch, for the reasons we stated.  The Type II functional response equation (aka the Holling Disk equation) has a long history of use in ecology for modeling the response of a population to changes in the availability of their food supply.  Had we used the Holling Disk equation to model the effects of catch, we would have: (1) encountered immediate difficulties in converting catches to changes in biomass, for example by using the biomass reduced by catch as the perturbation metric, because estimates of pollock biomass within our fur seal foraging area are not available to us; (2) considerably reduced the sensitivity of our analysis, because catches are usually less than 20% of biomass; and (3) used a functional form that is ill-suited to reflect the effects of fishing activity associated with catches on pup survival that are not primarily mediated by effects on pollock biomass.

The more critical point of comparison between our catch and biomass driven models involves imposing the same Leslie matrix projection of effects of either perturbation on pup births in subsequent years.  This insures that both models are on an equal footing with respect to the fur seal population response data, which we believe is far more important than any possible artifacts that may have been introduced by using functional forms relating catch or biomass to month 5 pup survival that are tailored to the particular perturbation (i.e. catch or biomass) involved. We therefore believe we made Models 1 and 2 as closely comparable as possible.

More generally, we note that no one has been able to find a correlative functional relationship between pollock biomass (or abundance) and fur seal survivals since inception of the modern EBS pollock fishery in the 1960s despite considerable effort to do so.  Recognition of this led us to look more closely at catch per se as a possible driver, and we were frankly surprised at how well catch could account for pup survival after accounting for the delay in detecting changes in pup survivals.  Perhaps our work will inspire others to discover equally strong correlations that are driven by changes in abundance, but we have not been able to do so.  We therefore respectfully decline to alter our manuscript in response to this comment.

  1. 9 line 316: " Assuming the area inhabited by the pollock population estimated in the pollock stock assessment remains nearly constant,"

Reviewer comment: "But pollock distribution is influenced by environmental conditions so it seems like this assumption would likely be violated. Not only might the total area change but the distribution of pollock in space within that area is also likely to change. How would violation of this assumption impact your interpretation?"

Response:

We agree that the pollock distribution is influenced by environmental conditions, and that both the total area occupied by the population varies interannually, as does the distribution of pollock in space.  Over the course of the pollock fishery data record from 1964 through 2018, pollock biomass has varied by a factor of more than 7.  Recognizing that primary productivity in the EBS is driven primarily by upwelling associated with the shelf break current, we doubt that the area inhabited by 90% of the EBS pollock stock varies by much more than ~±50%.  The distribution of pollock within this area is largely irrelevant, because By is a population average, and both fur seals and the pollock fishery are capable of searching across several hundred thousands of km2 for pollock aggregations.  However, if this assumption is substantially violated, then By may be interpreted as an index of relative total biomass.  We added a sentence here indicating this.

  1. 12 line 418: "Fig. 4B"

Reviewer comment: "Two things:

1) This may be a silly question but since you are talking about correlations, shouldn't the line in B be a linear fit?

2) All of the years since the ongoing decline (~1995 onwards or so) are well below the line and if you focused just on them they don't seem to fit any particular pattern with respect to pup production and pollock catches. While you could argue that you need those outliers (the early years, 64-66) to see the overall trend, I find it suspicious that the years during the ongoing decline don't fit at all. To me, at the very least this would seem to suggest that there may be slightly different mechanisms driving these different periods of decline or there is some other factor interacting here with this relationship and the ongoing decline that has not been quantified. This might need some additional exploration and certainly some discussion"

Response:

1)  The line in B is a LOWESS smoothing curve, which is helpful in showing where regression patterns change.  We did not indicate this, but have added this to the legend for Fig. 4B.  We only wanted to use correlation coefficients to scan the data for relationships. Correlations are a quick and easy way to look for a linear component of a relationship. However, the actual data may show much more complex relationships. The LOWESS smooth is be better suited to show the actual relationship.

The ideas behind this LOWESS smooths goes back to work at Bell Labs in the early 1980s, and can be found in papers and books by such authors as John Chambers, Beat Kleiner, and others (e.g., Graphical Methods for Data Analysis. 1983. Chambers, Cleveland, Kleiner, Tukey). More recently, others have tried to explain the advantage of this approach (e.g.., Loess: a nonparametric, graphical tool for depicting relationships between variables. W.G. Jacoby. Electoral Studies 19 (2000) 577–613), or Anderson (i.e., Nonparametric Methods for Modeling Nonlinearity in Regression Analysis, Robert Andersen. Annual Review of Sociology (2009) 35:1, 67-85). The important point is that we didn’t know ahead of time if these relationships were linear over the whole range or not, and of course, they can’t be linear over all possible values of seal population size.

2)  We thank this reviewer for pointing out that we were unclear on this point. Indeed, this reviewer’s observation is correct that there is not a simple linear relationship between pollock catch and pup births over all values of pollock catch.

Our operating hypothesis is that pup births are a function of breeding stock size together with pup survival and that the pollock catches influenced pup survival. In other words, when pollock catches are sustained at fairly high levels, under our operating hypothesis, the pup births should show a sustained decrease—especially after enough time has elapsed for the decrease in survival to sustain a decrease in the breeding population size. That is exactly what is shown in the figure in question: a sustained decrease in pup births from after the 1988 pollock harvest on, as pollock catches remained high. That seems to be exactly what the reviewer is pointing out. We had a hard time responding to this criticism because that is exactly what we wanted the reader to see.

From Figure 4B the reader should see a sustained decline in pup births starting earlier than 1995. Beginning about 1988 pollock catch, if there was a persistent reduction in pup survival because of something to do with the pollock catch, the peak response in pup births would be further delayed because a decrease in pup survival causes a delayed decrease in breeding stock size, as quantified by the Leslie matrix. Assuming that the operating hypothesis is correct, this decrease in breeding population size would cause a further decrease in pup births for the same level of pollock catch, which, in turn will further decrease the breeding population size.

Also, in the early 1970s there was a period of high pollock catch, but this was followed by a period of reduced pollock catch from about 1977 to 1987. Turning to Figure 4B, but without overanalyzing every single data point, we agree, there does seem to be a different reaction from the population when pollock catches are sustained at high levels from 1988 on. That seems to be what the reviewer noticed—a sustained decline in pup births lagged after 1988 or so—not the same linear relationship that was observed when pollock catches were low, and not the same reaction from the 1970s when high catches were mixed with lower pollock catch levels. That is our point.

We added text at lines 404 – 406 to hopefully make these points more clearly (and succinctly) in the manuscript.

  1. 13 line 435: "Fig. 5B"

Reviewer comment: "Two things:

1) This may be a silly question but since you are talking about correlations, shouldn't the line in B be a linear fit?

2) All of the years since the ongoing decline (~1995 onwards or so) are well below the line and if you focused just on them they don't seem to fit any particular pattern with respect to pup production and pollock catches. While you could argue that you need those outliers (the early years, 64-66) to see the overall trend, I find it suspicious that the years during the ongoing decline don't fit at all. To me, at the very least this would seem to suggest that there may be slightly different mechanisms driving these different periods of decline or there is some other factor interacting here with this relationship and the ongoing decline that has not been quantified. This might need some additional exploration and certainly some discussion"

Response:

Same as above.

  1. 15 line 491: "first-year survival has been affected by the magnitude of the pollock catch in the EBS."

Reviewer comment: " I think this statement needs to be softened here, since what the authors have shown is that there is a correlation between fishery catches and pup recruitment, which does not in and of itself prove that the fishery has contributed to the decline. It is plausible that fishery catches is picking up something else that is biologically important to fur seals, like pollock availability in general. The authors have not really addressed this nor provided enough information to allow this to be critically assessed. For example, I was left wondering what is it that drives interannual variability in fishing catch within the fur seal foraging areas - is it the distribution of fish? The explanation as to why the pollock biomass data don't fit as well was a bit unsatisfying and I don't think particularly well justified.

Response:

This criticism is really hard to respond to. It appears that what the reviewer objects to is just the very statement of the hypothesis. The statement in question is not a conclusion. It is a hypothesis.  Evaluation of this hypothesis is central to addressing the objective of this manuscript, as stated at line 96: " Our objective here is to evaluate the strength of any relationship between catches of 96 pollock by the EBS commercial fishery and estimates of Pribilof fur seal pup births after 97 explicitly accounting for changes in fur seal vital rates."

In the larger context of the whole paragraph, there was a conclusion. Perhaps we misunderstand the criticism, and perhaps the reviewer objected to the conclusion that the evidence “is supportive of” our hypothesis. We did not say that we conclusively showed that the fishery is responsible for all of the seal population decline. And we do not know why the pollock biomass data do not fit better. Maybe someone else can explain this, or maybe not. We have simply reported what we found. Either way, we do think that we are fully justified in concluding that the evidence is “supportive of” our stated hypothesis for all the reasons we have explained in the discussion.

As for the plausibility that fishery catches are picking up something else that is biologically important to fur seals, like pollock availability in general, we believe this is doubtful.  As we noted at lines 715 – 716, exploitation rates of the pollock fishery have mostly remained below 20% since the 1980s.  These exploitation rates often approach 10% during this period.  These low exploitation rates have been routinely cited by the industry and by the North Pacific Fisheries Management Council (NPFMC), the body that sets catch limits for the EBS pollock fishery, as evidence in support of the hypothesis that commercial fishing has had no effect on the fur seal population.  Perhaps the effective exploitation rates within our fur seal foraging area are higher, but pollock biomass estimates within this area are not available to us. Evaluation of this hypothesis would require the NMFS to conduct the analysis.

Regarding the drivers of interannual variability in fishing catch within the fur seal foraging area, the over-arching driver is the catch limits set by the NPFMC.  As illustrated in our Fig. 2B, 78% of the catch on average comes from within our fur seal foraging area.  We added a new paragraph at line 630 describing the historical and current management constraints on pollock catches in more detail.

As far as the question of why the biomass does not fit better, we don’t know why, and don’t see a lot of reason to speculate about this more than we have. We have simply reported the result we obtained. 

  1. 16 line 520: "2000 to 2002, and 2014 to 2018"

Reviewer comment: " The recent decline at St. Paul has really being going on since the late 1990s so this designation here is not really correct"

Response:

We thank this reviewer for bringing this error to our attention.  The "2000 to 2002" should have been "2000 to 2010".  Inspection of the pup birth estimates (see our Fig. 2A) shows that the decline appears to have been temporarily arrested from 2010 to 2014.

  1. 16 line 525: "...pollock biomass..."

Reviewer comment: "From my understanding though these do not appear to be exactly equivalent comparisons, as pollock biomass is across the entire EBS and is age 1+, correct? Because of this I don't think you can really confidently say that a more precise measure of biomass would not improve the relationship"

Response:

We attempted to address this comment by conducting separate Model 2 analyses based on total age-1+ biomass and total ages 1 –3 biomass. We agree that it would be desirable to base our Model 2 analyses on metrics of pollock biomass specific to our fur seal foraging area, but we do not have ready access to these data. We could make another data request to the NMFS, but based on the time it took the NMFS to respond to our request for pollock catch data within our fur seal foraging area, it could take a year or more to receive the results, assuming NMFS agreed to furnish them.  They might reasonably refuse to agree, because we would in effect be asking them to conduct a separate assessment for the portion of the pollock stock within our fur seal foraging area, and the agency may not have the resources, or assign sufficient priority to the considerable effort this would entail.

We certainly never did say that a more precise measure of biomass would not improve the relationship. It certainly may—we agree. Or it may not. Since nobody that we know of has a more precise measurement, we cannot know. Also, this is really a question of accuracy more than precision.  The accuracy of the EBS pollock stock assessments declines the farther into the past one looks, and there is no way to improve on this.  So while we would like to conduct our biomass models on the basis of more accurate or precise data specific to our fur seal foraging area, we don't see how we can do this.

  1. 17 line 548: "...abundance, suggests an artifact perhaps resulting from the relatively high auto-correlation in the pollock abundance and pup birth time series."

Reviewer comment: "I think it is hard to argue here that the auto-correlation in the datasets is producing a spurious result here and not expect it to impact the results of the other analysis simply because that analysis provides a trend that was expected"

Response:

We simply used correlation as a tool to scan for relationships. We are not trying to make these correlations out to mean more than what they actually mean. We earlier cautioned that autocorrelation can make two autocorrelated time series appear to be more related than they actually are. We cited Pyper et al. on line 263 to support this point.  There is a secular trend in the pup births series. Therefore, any series with a slight trend it will show an elevated correlation with the pup birth series at lag zero and possibly other lags. Indeed, both pollock catch and pollock biomass measures have trends in them. As the reader evaluates these cross correlations, this needs to be kept in mind.

If a cross correlation peaks at some value, then that is only a clue—not a confirmation of a hypothesis. It does not lead to a solid confirmation of cause and effect. We are only trying show and report what the correlations are, and then trying to put the clues together. The correlation of the pup births with biomass peaks at lag zero. That does not fit with any biological hypothesis we are aware of and it is easy to see how this could be caused by the secular trend in the pup birth series. Maybe someone else can will get some insight from that observation, or maybe not. We provided Figure 4B and 5B so the reader can examine the relationships for the years with the peak correlation for him or herself.

A more striking observation is that the dominate peak in correlation between pollock catch and pup births occurs at lag -11. Again, we provided Figure 4B so the reader can inspect the actual pattern in the data for him- or herself. It may well be that some of this correlation is due to autocorrelation. That is not an unreasonable conclusion. But the fact that the dominate peak is at lag -11 certainly seems us to be an important clue.

  1. 17 line 572: "...but is sensitive to factors associated with pollock catches."

Reviewer comment: "To me this doesn't really make sense - perhaps if you elaborate on some of these factors associated with pollock catches that would be helpful. In general, I found myself wanting more information on what exactly is driving the interannual variation in catch. This result also wouldn't be particularly surprising if you are using pollock biomass across the entire EBS, since then yearly values may not reflect the conditions that fur seals experienced in their foraging areas. You also discuss in this paragraph that there is a lot of uncertainty in pollock abundance in early years, which when you look at Figure 4, are the years that seem to be driving this relationship. If those points are turned from "low biomass-high pup production" to "high biomass-high pup production" then it seems you probably need to re-run the analysis (not just state the it weakens the correlation)."

Response:

By “factors associated with pollock catch,” we simply meant pollock catch or any other unobserved phenomenon—a phenomenon we do not have measurements for—that is related to the pollock harvest and could affect pup survival. The proposed re-analysis is just outside the scope of this paper. What we have shown is that pup births seem to be related to pollock catch several years in the past, like you would expect if pollock catch affected pup survival. That is the conclusion we have come to from the data we actually have.

It is not our intention to keep analyzing until we find a way to show a stronger pollock abundance relationship with pup births. Nor is it our intention justify why we could not find a stronger relationship. Maybe it seems like those pup births should be more related to pollock biomass measures, but we have simply reported the correlations we observed using available data. Further, we did offer some speculation about this data, but we have nothing further to add. To expand analyses of the biomass data and why there is not a stronger relationship is simply outside of the scope of our paper, as stated in the first sentence of the last paragraph of the introduction.

Regarding discussion of what is driving interannual variation in pollock catches, see the new paragraph we added at line 630 describing the historical and current management constraints on pollock catches in more detail.

  1. 17 line 577: "...high-density aggregations of pollock"

Reviewer comment: "There should be some discussion somewhere that fur seals eat young pollock that is not the target of the fishery, particularly in years where there is abundant young pollock."

Response:

We added a new subsection to the Discussion that addresses this issue at line 687.

  1. 18 line 610: "Carrying capacity"

Reviewer comment: " While I don't disagree with the logic here about carrying capacity, following this logic it seems you should have detected a relationship between pollock biomass as well, since fishery catches aren't necessarily indicative of the available prey resources. Somewhere in the discussion I think it also has to be raised that while lactating females eat large pollock, they also eat a lot of pollock of size classes not targeted by the fishery"

Response:

We do not agree that the logic presented in this paragraph regarding the EBS carrying capacity for fur seals necessarily implies that we should have detected a relationship between fur seal recruitment and pollock biomass.  We stated that the declining trajectory of the fur seal heard "...suggests that there has been a dramatic reduction in the ability of the EBS to support northern fur seal breeding requirements," (emphasis added).  This does not necessarily imply a reduction in pollock (or other ) prey biomass.  It could also refer to other factors, and the factor we suspect most is the disruption of large aggregations of pollock by the fishery.  We agree that fishery catches aren't necessarily indicative of available prey resources.  But given the low exploitation rates of the EBS pollock fishery, we find it difficult to account for the sustained decline of the fur seal herd over the course of several decades on account of a reduction of pollock availability to 80% – 90% of the unfished biomass.

Regarding discussion of pollock size selectivity by fur seals, see our response to the previous comment.

  1. 19 line 696: " The continuing displacement of northern fur seals from the Pribilofs to Bogoslof Island, during a period when the herd has been in a state of overall decline, suggests competitive displacement of Pribilof northern fur seals in response to competition with the pollock fishery,"

Reviewer comment: "I think this statement really oversteps the bounds of the data. First, the authors don't provide any data for how much emigration did or is currently occurring from the Pribs to Bogoslof. While of course there must have been some initially, just because the population is growing doesn't mean that the growth now isn't due to animals originating from Bogoslof, particularly as adult females exhibit high site fidelity to rookeries. Second, there is no evidence to suggest the cause of emigration. While the timing is roughly coincident with the larger decline, it is possible that the colonization of Bogoslof happened by chance and that the simple presence of animals attracted others. Colonization of new areas (or previously extirpated areas) happens under growing populations as well, and in the 1960s and 1970s northern fur seals also began to repoulate sites within their historical range, such as on the Farallon Islands"

Response:

We agree that the cause of the emigration of fur seals from the Pribilofs to Bogoslof Island is conjectural, and have modified the text to reflect this more clearly. We also added references that support substantial immigration from the Pribilofs through at least the late 1990s.

Reviewer 2 Report

This is an important ms that describes the decline of the northern fur seal population relative to the pollock fishery. This is a very dense paper, with a lot of information. However, it is well presented and the conclusions are carefully developed. I particularly appreciated the discussion concerning how the fishery impacts fur seal foraging by disrupting large aggregations. As the authors point out large vertebrate predators rely on large dense aggregations of prey. Having a lot of prey is not the same thing as high prey availability that is associated with highly aggregated prey swarms. My only suggestion is they consider adding a reference to lines 590-591. In this section, they refer to how longer trips reduce milk intake to fur seal pups. 

Costa, D. P. 2008. A conceptual model of the variation in parental attendance in response to environmental fluctuation: foraging energetics of lactating sea lions and fur seals. Aquatic Conservation: Marine and Freshwater Ecosystems 17:S44-S52.

The above paper specifically describes this phenomenon with reference to fur seals in general, but also with respect to northern fur seal in particular.

Author Response

Reviewer #2:

We added the reference suggested by this reviewer, and we are grateful that this was pointed out to us.

Reviewer 3 Report

The manuscript provides an important contribution on the population dynamics and trends of the northern fur seal in context of the pollock fisheries in the Eastern Bering Sea. I include only a few minor comments for revision and to improve clarity of the manuscript, but otherwise recommend publication.

I would have also appreciated some additional discussion of the changing trends in the fisheries catch data for the eastern Bering Sea, and how that may affect predators in the region to put this study in a broader context of changing fisheries in the sub-Arctic. However, I would not consider such additions necessary to warrant publication of this manuscript.

Author Response

Reviewer #3

Specific Comments:

  1. 2 line 72: Define "B" season.

Response:

We replaced "B" with "fishing" here, where the fishing period involved is stated immediately before this in the same clause.

  1. 3 lines 101 - 107: The following text for Method 2 is too long and complicated.

Response: We simplified this considerably.

  1. 12 lines 425 - 432: Change "abundance" to "biomass" for consistency.

Response: We made this change throughout the manuscript.

Fig. 1C: Caption should also explain what the colors refer to in Figure 1C.

Response:  We added this explanation to the caption of this figure.

Fig. 5B: Figure caption is not consistent with figure axis label.

Response:  We fixed this for figures 4A, 4B, 5A and 5B.

Round 2

Reviewer 1 Report

I appreciate the time the authors invested in addressing my comments. I just have three minor responses to their responses.

Original comment from Line 174 regarding the different population trends: Yes, the number is irrefutable but it seems appropriate to give the readers all the information. To that effect, it seems like a small thing to simply add a statement similar to what the authors had in their response, that the broad trends across the past five decades are similar, although in the past two decades there is evidence that the population on St. George may have stabilized.

Original comment on Line 307: The authors response satisfies my comment but I think it would be beneficial to add a statement in this section that summarizes their response, as there might be other readers that have the same concerns I initially did. That is, a simple 1 or 2 sentences highlighting that the models indeed are comparable would be valuable.

Comments regarding the issue of biomass vs catch data: The responses generally satisfied my concerns, but since the biomass data are not foraging-range specific, I would still suggest reminding the reader of this in the discussion. This could be very small, for example by clarifying in the statement on Lines 719 -721 that results are insensitive to eastern Bering Sea pollock biomass (i.e. inside and outside the foraging range of lactating females).